# Loss of 5-methylcytosine alters the biogenesis of vault-derived small RNAs to coordinate epidermal differentiation

Abdulrahim A. Sajini[1,2,3], Nila Roy Choudhury[4], Rebecca E. Wagner[1], Susanne Bornelöv [5], Tommaso Selmi[1], Christos Spanos [6], Sabine Dietmann[5], Juri Rappsilber[6,7], Gracjan Michlewski [4,6,8] & Michaela Frye[1,9]

The presence and absence of RNA modifications regulates RNA metabolism by modulating the binding of writer, reader, and eraser proteins. For 5-methylcytosine ($m^5C$) however, it is largely unknown how it recruits or repels RNA-binding proteins. Here, we decipher the consequences of $m^5C$ deposition into the abundant non-coding vault RNA VTRNA1.1. Methylation of cytosine 69 in VTRNA1.1 occurs frequently in human cells, is exclusively mediated by NSUN2, and determines the processing of VTRNA1.1 into small-vault RNAs (svRNAs). We identify the serine/arginine rich splicing factor 2 (SRSF2) as a novel VTRNA1.1-binding protein that counteracts VTRNA1.1 processing by binding the non-methylated form with higher affinity. Both NSUN2 and SRSF2 orchestrate the production of distinct svRNAs. Finally, we discover a functional role of svRNAs in regulating the epidermal differentiation programme. Thus, our data reveal a direct role for $m^5C$ in the processing of VTRNA1.1 that involves SRSF2 and is crucial for efficient cellular differentiation.

[1] Department of Genetics, University of Cambridge, Downing Street, Cambridge CB2 3EH, UK. [2] Department of Biomedical Engineering, Khalifa University of Science and Technology, P.O. Box 127788, Abu Dhabi, United Arab Emirates. [3] Department of Medical Laboratory Technology, University of Tabuk, Tabuk, P.O. Box 71491, Saudi Arabia. [4] Division of Infection and Pathway Medicine, University of Edinburgh, The Chancellor's Building, 49 Little France Crescent, Edinburgh EH16 4SB, UK. [5] Wellcome MRC Cambridge Stem Cell Institute, Tennis Court Road, Cambridge CB2 1QR, UK. [6] Wellcome Centre for Cell Biology, University of Edinburgh, Michael Swann Building, Max Born Crescent, Edinburgh EH9 3BF, UK. [7] Department of Biotechnology, Technische Universität Berlin, Gustav-Meyer-Allee 25, 13355 Berlin, Germany. [8] ZJU-UoE Institute, Zhejiang University, 718 East Haizhou Road, Haining, Zhejiang 314400, P.R. China. [9] German Cancer Research Centre (Deutsches Krebsforschungszentrum, DKFZ), Im Neuenheimer Feld 280, 69120 Heidelberg, Germany. Correspondence and requests for materials should be addressed to G.M. (email: Gracjan.Michlewski@ed.ac.uk) or to M.F. (email: M.Frye@dkfz.de)

The post-transcriptional deposition of chemical modifications into RNA emerged as a crucial regulator of gene expression programs[1]. 5-methylcytosine (m5C) occurs in various RNA molecules and is mediated by at least eight, highly conserved enzymes (NSUN1-7, and DNMT2) in mammals[2]. One of the best-characterised m5C methyltransferase is NSUN2, which targets the majority of cytoplasmic transfer RNAs (tRNAs) and a smaller number of coding and other non-coding RNAs including VTRNAs[3–7].

In tRNAs, NSUN2-mediated formation of m5C protects from endonucleolytic cleavage[7,8]. Loss of tRNA methylation enhances the affinity to the endonuclease angiogenin, which then cleaves the tRNAs causing a global reduction in protein synthesis[7,8]. The cellular consequences of NSUN2-deletion are reduced cellular migration and delayed activation of stem cell differentiation[8–10]. NSUN2-mediated RNA methylation is required for normal development[7], and loss-of-function mutations in human NSUN2 gene is associated with neuro-developmental disorders[11–14].

The functional role of m5C in VTRNAs is less clear. VTRNAs are integral components of large ribonucleoprotein vault particles found in the cytoplasm of most eukaryotic cells[15,16]. However, only about 5% of cytoplasmic VTRNA is directly associated to vault particles and similarly small amounts of VTRNAs are reported to reside in the nucleus[17,18]. In humans, four VTRNAs are expressed VTRNA1.1, VTRNA1.2, VTRNA1.3, and VTRNA2.1[16], two of which (VTRNA1.1 and VTRNA1.3) are methylated by NSUN2[3]. VTRNAs have been implicated in the cellular immune response, cell survival and oncogenic multi-drug resistance, indicating a functional role in several fundamental biological processes[17,19–23].

VTRNAs are also processed into smaller regulatory RNAs (svRNA) by a pathway different from microRNA (miRNA) biogenesis[21]. VTRNA-derived svRNAs are highly abundant in exosomes, and at least some of them regulate gene expression similarly to miRNAs[3,21,24,25]. We previously revealed that the processing of full-length VTRNA1.1 into svRNAs depended on the methylation of cytosine 69 (C69)[3], yet the underlying molecular mechanisms and the functional role of the svRNAs remained unknown.

Here, we performed mass spectrometry-based quantitative proteomics to identify all proteins whose binding affinity is directly determined by the presence or absence of m5C69 in VTRNA1.1. We identify SRSF2 as a novel VTRNA-binding protein that is repelled by m5C69. By binding the un-methylated form with higher affinity, SRSF2 protects VTRNA1.1 from processing. We confirm that both NSUN2 and SRSF2 coordinate the processing of VTRNA1.1 into specific svRNAs. Functionally, we show that the presence of one specific VTRNA-derived small non-coding RNA (svRNA4) is sufficient to alter the transcriptional program needed to induce epidermal differentiation. Together, we demonstrate that the deposition of m5C orchestrates VTRNA1.1 processing and thereby determines its downstream biological function.

## Results

**Methylation of VTRNA1.1 requires NSUN2.** NSUN2 methylates the vast majority of tRNAs and a small number of coding and non-coding RNAs[1]. To determine which of these methylated sites solely depended on NSUN2, we rescued human dermal fibroblasts lacking a functional NSUN2 protein (NSUN2−/−) by re-expressing NSUN2 or an enzymatic dead version of the enzyme (K190M)[11,26]. We confirmed cytosine- methylation using RNA bisulfite (BS) sequencing[7]. As expected, the methylation levels between NSUN2−/− cells infected with the empty vector control and the enzymatic dead NSUN2 (K190M) highly

correlated (Fig. 1a). In contrast, methylation of more than 100 sites significantly increased when NSUN2 was re-expressed (Fig. 1b; Supplementary Data 1). We confirmed that the NSUN2 and K190M proteins were equally expressed in the rescued NSUN2−/− cells (Supplementary Fig. 1a).

In addition to tRNAs, we confirmed NSUN2-specific methylation of cytosine (C) 69 in VTRNA1.1 (Fig. 1c)[3]. Furthermore, we identified a small number of novel high-confidence sites in both coding and non-coding RNAs (Supplementary Fig. 1b-e). NSUN2-dependent methylation sites in VTRNAs, RPPH1, and HECTD1 for instance, were consistently methylated in human cells, including HEK293 and human embryonic stem cells (H9) (Fig. 1d, e). Notably, the methylation levels of C69 in VTRNA1.1 varied and were usually lower than 50%, even when NSUN2 was over-expressed (Fig. 1c–e; left hand panels). We concluded that methylation of VTRNA1.1 at C69 occurred at dynamic levels but was widely present in human cells.

**VTRNA1.1 methylation determines the biogenesis of svRNA4.** The functional relevance of C69 methylation in VTRNA1.1 was unknown. However, our previous study demonstrated that the presence and absence of m5C correlated with the differential processing of VTRNA1.1 into small non-coding RNA fragments (svRNA1-4) (Supplementary Fig. 1f)[3]. Of these svRNAs, only the length of svRNA4 coincided with C69 (Fig. 1f; Supplementary Fig. 1f). SvRNA4 was more abundant in human dermal fibroblasts expressing NSUN2 (Fig. 1g), indicating that the formation of svRNA4 was enhanced when VTRNA1.1 carried m5C69[11]. We asked whether the presence of m5C69 increased the formation of svRNA4 and quantified svRNA4 in NSUN2−/− cells re-expressing the wild-type (wt) or enzymatic dead versions of NSUN2 (C321A; C271A)[8,26]. The processing of VTRNA1.1. into svRNA4 depended on the methylation activity of NSUN2 because only the wild-type construct of NSUN2 increased svRNA4 production (Fig. 1g). All over-expressed constructs were equally up-regulated in the NSUN2−/− cells (Fig. 1h)[8]. Thus, the presence of a methylation group at C69 enhanced the processing of VTRN1.1 into svRNA4.

**Proteins binding to un-methylated and methylated VTRNA1.1.** To dissect how VTRNA1.1 processing was regulated, we sought to identify all RNA-binding proteins showing a higher affinity to methylated or un-methylated VTRNA1.1. We performed quantitative RP-SMS (RNA pull-down SILAC (stable isotope labeling with amino acids in cell culture) mass spectrometry) in two independent experiments (Supplementary Fig. 2a; Supplementary Data 2 and 3)[27]. We found a high correlation of identified proteins between the technical replicates (Supplementary Fig. 2b) and identified a total of 144 proteins commonly bound to VTRNA1.1 in two independent experiments (Fig. 2a; Supplementary Fig. 2c). Gene Ontology (GO) analyses confirmed that we significantly enriched for proteins binding to single and double stranded RNAs (Fig. 2b; Supplementary Data 4).

As expected, the majority of RNA-binding proteins were not differentially bound to methylated or un-methylated VTRNA1.1, this included proteins related to m5C deposition (NSUN2) and 'reading' (ALYREF) (Fig. 2a)[4]. In contrast, pseudouridine synthase 7 (PUS7) bound methylated VTRNA1.1 with higher affinity, whereas the serine/arginine rich (SR) splicing factor 2 (SRSF2) was consistently repelled by m5C in VTRNA1.1 (Fig. 2a). To confirm that we pulled down VTRNA1.1-specific RNA-binding proteins, we performed RNA pull-down assays followed by western blot, and found that SRSF2 bound non-methylated (C69) with higher affinity than methylated (m5C69) VTRNA1.1 (Fig. 2c). In contrast, PUS7 bound methylated VTRNA1.1 with

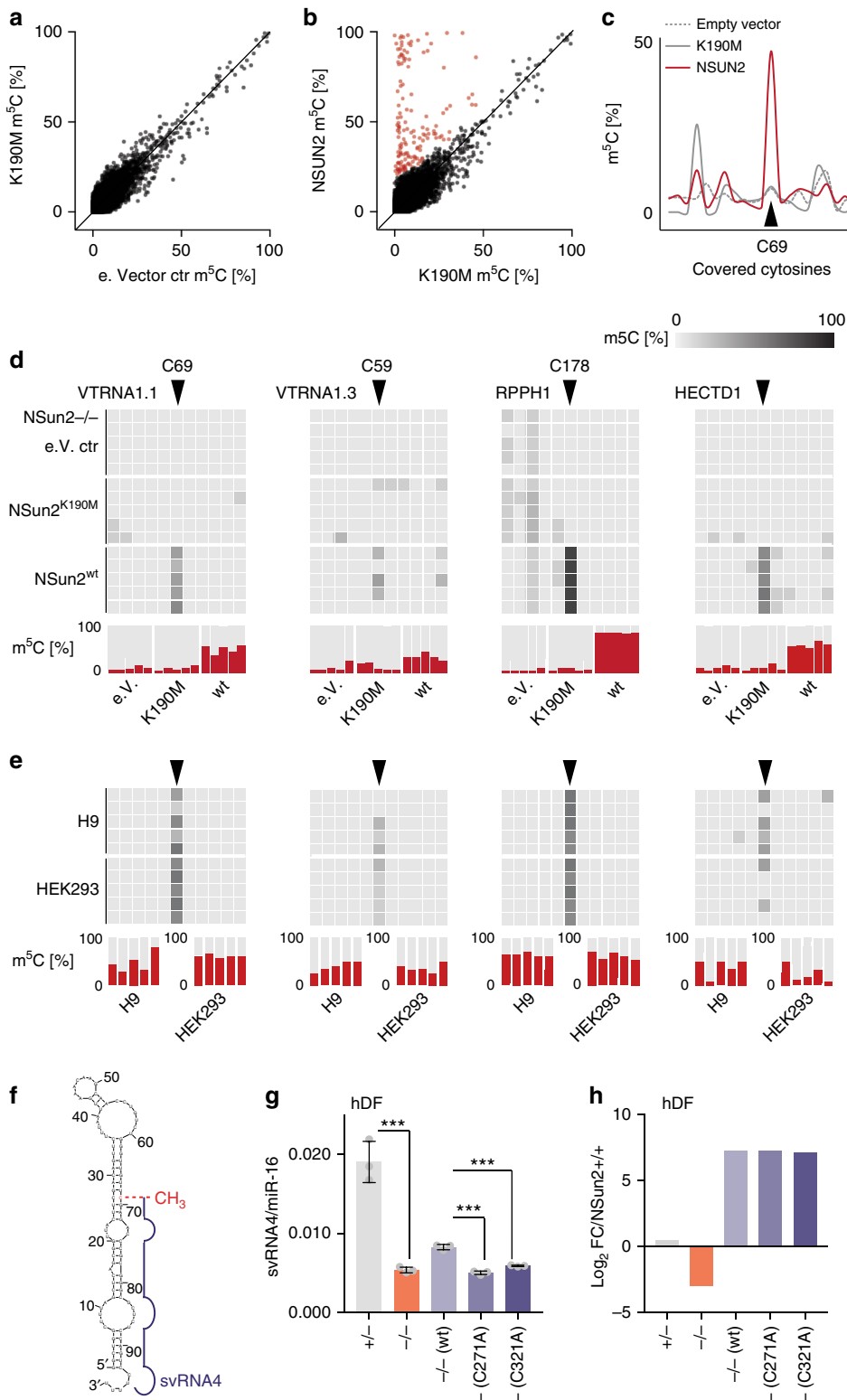

**Fig. 1** Methylation of VTRNA1.1 by NSUN2 determines the biogenesis of svRNA4. **a–c** Correlation of site-specific methylation (m5C) levels (**a**, **b**) and methylation level at all covered cytosines in VTRNA1.1 (**c**) in *NSUN2-/-* cells infected with the empty (e.) vector (ctr), the enzymatic dead construct K190M or the wild-type NSUN2 construct. NSUN2-specific sites are highlighted in red. **d**, **e** Heatmaps (upper panels) and methylation level (bottom panels) of VTRNA1.1, VTRN1.3, RPPH1, and HECTD1 in infected *NSUN2-/-* cells (**d**), human embryonic fibroblasts (H9) and HEK293 cells (**e**). Shown are five independent bisulfite conversion experiments. **f** Schematic illustration of NSUN2-dependent methylation (CH3) of VTRNA1.1 and the small regulatory non-coding fragments svRNA4. **g** Abundance of svRNA4 in the presence (+/−) and absence (−/−) of NSUN2. Methylation dead NSUN2-mutant constructs (C271A; C321A) fail to rescue svRNA4 levels in *NSUN2−/−* cells. Error bars indicate s.d. ($n = 3$ qRT-PCR reactions). ***$p < 0.001$ unpaired student's *t*-test. **h** Log2 fold-change of *NSUN2* in the indicated cells compared to *NSUN2+/+* control cells measured by ribosome profiling. Source data are provided as a Source Data file

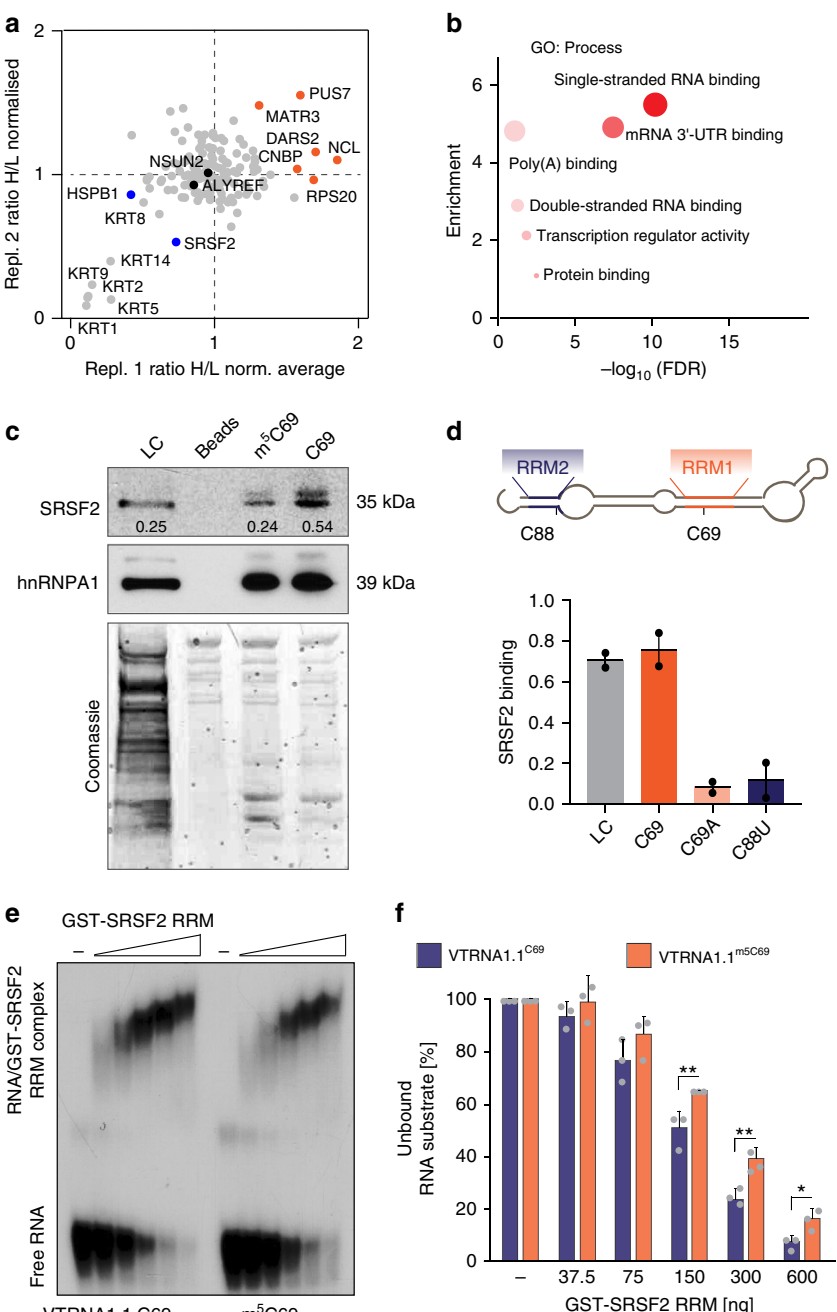

**Fig. 2** SRSF2 preferentially binds un-methylated human VTRNA1.1. **a** Of the 144 common proteins binding to VTRNA1.1 in two different RP-SMS experiments, a small number bound methylated (red) or unmethylated (blue) VTRNA1.1 with higher affinity. **b** Gene Ontology (GO) analyses of the 144 commonly bound proteins. **c** Western blot and Coomassie stain for SRSF2 in HeLa cell lysates pulled-down with agarose beads coupled to methylated (m5C69) or un-methylated (C69) Vault-RNA1.1 (upper panel). hnRNP A1 serves as a loading and RNA-binding control (lower panel). Numbers indicate band intensity vs. loading control. **d** Location of the putative SRSF2 RNA-binding motifs (RRM1 and RRM2) in VTRNA1.1 (upper panel) and RNA pulldowns using wildtype or mutated (C69A; C88U) VT-RNA1.1-constructs to confirm both putative SRSF2 binding sites are necessary for SRSF2 binding. Shown is mean and range (n = 2 independent experiments). Quantification in (**c**, **d**) was done using ImageJ. **e** EMSA assay using methylated (m5C69) and unmethylated (C69) VTRNA1.1 to measure binding of recombinant SRSF2. **f** Quantification of (**e**). Error bars indicate s.d. (n = 3 experiments). **p < 0.01, *p < 0.05 students t-test. Source data are provided as a Source Data file

higher affinity (Supplementary Fig. 2d; C69, m5C69). The heterogeneous nuclear RNP (hnRNP) A1 protein served as a loading control (Fig. 2c)[27].

**m5C directly influences the affinity of SRSF2 to VTRNA1.1.** SRSF2 is best-known for its role in splicing where it mediates

exon inclusion and exclusion equally well[28]. SRSF2 was a promising candidate to regulate VTRNA1.1 processing because SR protein binding is not limited to pre-mRNA, they can also associate with non-coding RNAs, such as 7SK and MALAT1[29–32]. SRSF2 binds to pre-mRNA via its RNA recognition motif domain[33–35], and VTRNA1.1 contained two putative SRSF2 RNA binding motifs (RRM1 and RRM2) (Fig. 2d; Supplementary

Fig. 2e)[28,36]. RRM1 overlapped with the methylated cytosine 69 (Fig. 2d). To validate the functional importance of the SRSF2 binding motifs, we point mutated C69 (C69A) and C88 (C88U) in VTRNA1.1 and performed RNA pull-down experiments followed by western blot for SRSF2. Both mutations decreased the binding affinity to SRSF2 (Fig. 2d; Supplementary Fig. 2f). Interestingly, binding of PUS7 to the mutated VTRNA1.1-constructs was also decreased (Supplementary Fig. 2d).

To further confirm a direct RNA–protein interaction between VTRNA1.1 and SRSF2, we performed electrophoretic mobility shift assays (EMSA) using methylated and un-methylated VTRNA1.1 (Fig. 2e). Purified, recombinant GST-SRSF2 RNA recognition motif (GST-SRSF2 RRM) showed high binding affinity to un-methylated VTRNA1.1 (C69) that correlated with increased concentration of the recombinant protein. Importantly, the binding affinity of GST-SRSF2 RRM was significantly lower when VTRNA1.1 was methylated at C69 ($m^5$C69) (Fig. 2e, f).

Finally, we asked whether SRSF2-binding affinity to VTRNA1.1 also decreased upon NSUN2-mediated methylation in vivo. First, we confirmed comparable protein expression levels of SRSF2 in NSUN2-expressing (+/+; +/−) or –lacking (−/−) cells by immunoprecipitation or western blotting (Fig. 3a, b; Supplementary Fig. 3a). Next, we co-immunoprecipitated SRSF2 and measured bound VTRNA1.1 by qRT-PCR (Fig. 3c). The amount of VTRNA1.1 bound to SRSF2 was highest in the absence of NSUN2 (Fig. 3c), confirming that SRSF2 preferentially bound un-methylated VTRNA1.1.

In conclusion, we identified SRSF2 as a novel VTRNA1.1-binding protein, whose affinity was reduced by post-transcriptional methylation of VTRNA1.1 by NSUN2.

**VTRNA1.1 processing is altered in the absence of SRSF2.** To test how SRSF2 modulated the processing of VTRNA1.1, we depleted SRSF2 in NSUN2-lacking cells for two reasons. First, the affinity of SRSF2 to VTRNA1.1 was highest in NSUN2−/− cells (Fig. 3c). Second svRNA4 production was lowest in NSUN2−/− cells (Fig. 1g). While SRSF1, a close protein family member of SRSF2, was efficiently down-regulated using siRNA and shRNAs, SRSF2 was consistently repressed by only 50% (Supplementary Fig. 3b, c). However, both proteins were similarly down-regulated on protein levels (Fig. 3d). The processing of VTRNA1.1 into svRNA4 was significantly increased when SRSF2 was knocked-down (Fig. 3e). Together, these results showed that (i) SRSF2 protected un-methylated VTRNA1.1 from processing and (ii) loss of SRSF2 was sufficient to rescue svRNA4 production in the absence of NSUN2-driven methylation. This data indicated that SRSF2-binding to VTRNA1.1 was likely to occur up-stream of VTRNA1.1 methylation. The processing into svRNA1 was slightly reduced by down-regulation of SRSF1 and 2, yet this change was not significant (Fig. 3e). In summary, our data demonstrated that SRSF2-binding to un-methylated VTRNA1.1 down-regulated the level of processing into svRNA4.

**VTRNA1.1, svRNA4 and SRSF2 levels are dynamically regulated.** We next asked whether SRSF2-regulated VTRNA1.1 processing was physiologically relevant. Because removal of SRSF2 can cause cell death[37], we turned our studies to primary human keratinocytes (HK), which are highly resistant to apoptosis[38]. Primary HK can be cultured as undifferentiated, lineage committed progenitor cells and induced to terminally differentiate by exposure to high calcium concentration in the culture medium[39]. Using the calcium switch assay, we differentiated epidermal cells for 2 and 6 days (Fig. 4a).

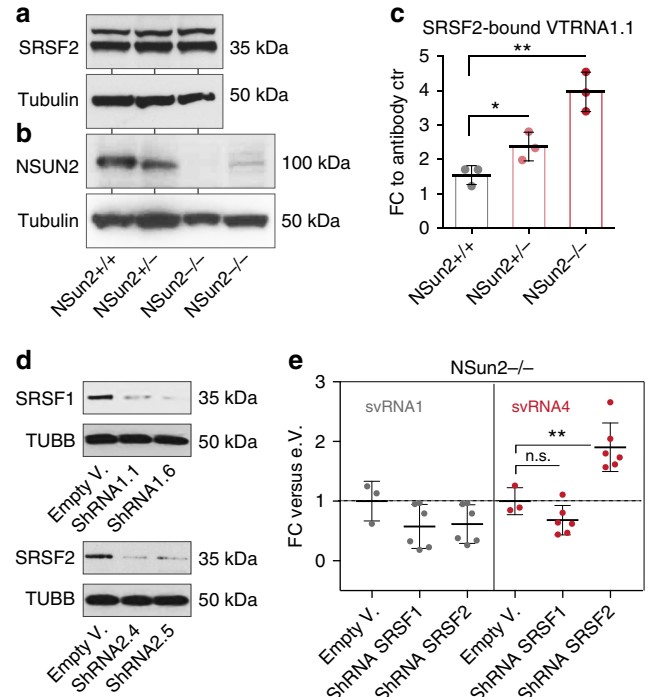

**Fig. 3** Methylation-guided VT-RNA1.1 processing is altered in the absence of SRSF2. **a, b** Western blot for endogenous SRSF2 (**a**) and NSUN2 (**b**) in NSUN2-expressing (+/+, +/−) and -lacking (−/−) human fibroblasts (from two patients). Tubulin served as a loading control. **c** qRT-PCR measuring SRSF2-bound VTRNA1.1 normalised to the control (ctr; Rabbit serum conjugated with Dynabeads) after recovering the pulled down RNA in the SRSF2 immunoprecipitation. **d** Western blot for SRSF1 and SRSF2 in NSUN2−/− human fibroblasts infected with shRNAs (1.1, 1.6, 2.4, 2.5). Tubulin served as a loading control. **e** Fold-change (FC) of svRNA1 and 4 abundances after knock-down of SRSF1 and 2, relative to NSUN2−/− cells infected with the empty vector (e.V.). Shown are the pooled values using the two shRNA constructs shown in (**d**). Error bars indicate s.d. (n = 3-6 qRT-PCRs). **p < 0.01 students t-test. Source data are provided as a Source Data file

First, we confirmed that RNA expression levels of the epidermal differentiation markers keratin 10 (Krt10), transglutaminase I (Tgm), involucrin (Inv), and the epidermal differentiation regulator Ovo Like Transcriptional Repressor 1 (Ovol1) were all up-regulated at both time points (Fig. 4b)[40,41]. Notably, also the abundance of full-length VTRNA1.1 significantly increased upon differentiation (Fig. 4b). In contrast, the RNA levels of Srsf2, Nsun2 and svRNA4 were all repressed upon differentiation (Fig. 4c). Together, these data indicated that both methylation of RNAs by NSUN2 and VTRNA1.1 processing was highest in undifferentiated epidermal progenitor cells.

**m5C levels are dynamic and depend on the type of RNA.** To directly determine the methylation differences in undifferentiated and differentiated epidermal cells, we performed RNA bisulfite (BS) sequencing[7,42]. We only quantified sites with a coverage of more than 100 reads and a minimum of 20% methylation in at least one of the two conditions (Supplementary Data 5). The average methylation level of the around 350 identified sites was significantly higher in the undifferentiated cells (Fig. 4d). We confirmed comparable coverage of these sites in the two conditions (Fig. 4e). Next, we compared the methylation levels of NSUN2-dependent and high confidence sites, which we defined as all rescued sites found in the NSUN2−/− human dermal

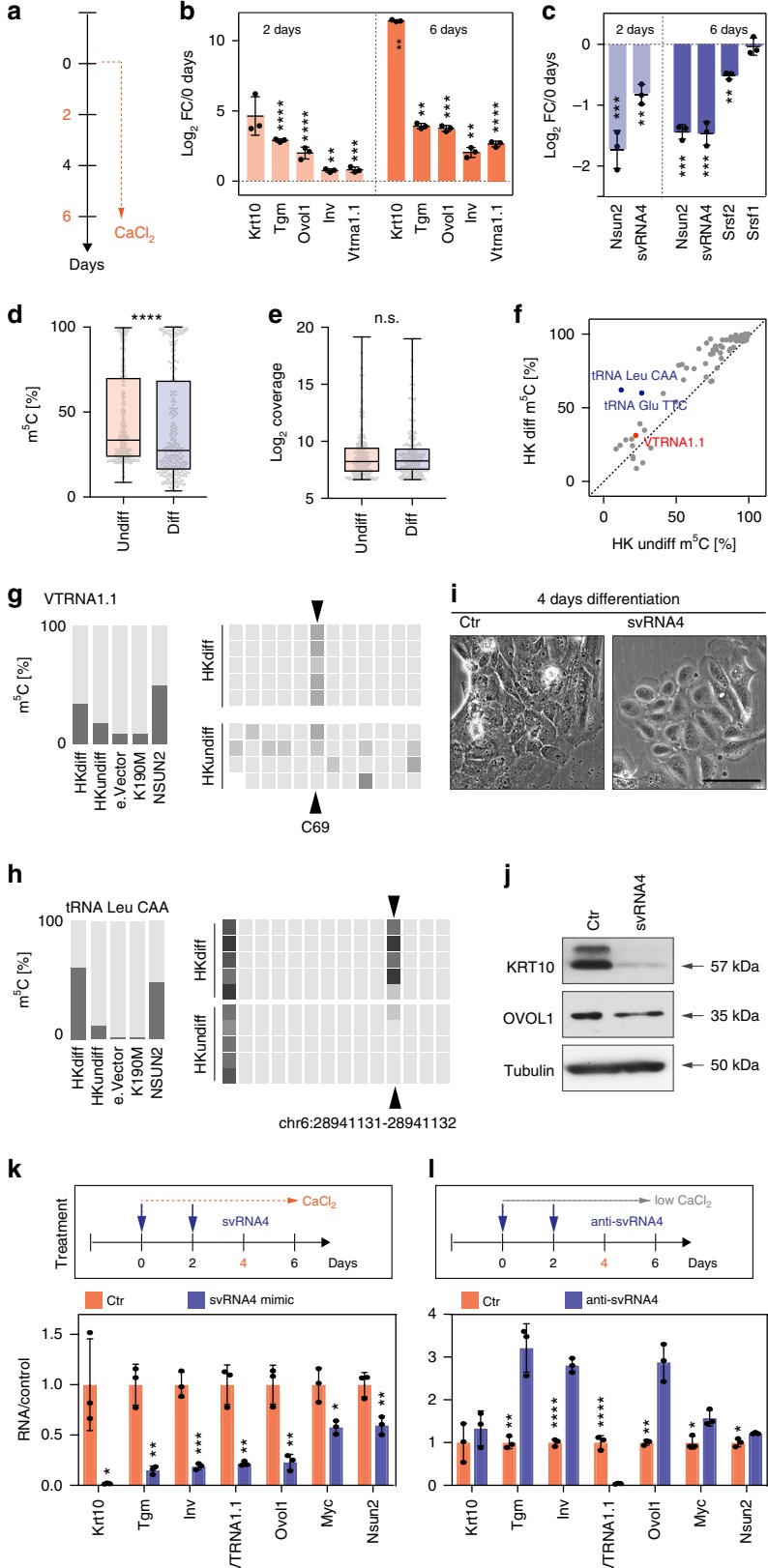

fibroblasts ($n = 76$) (Supplementary Data 6). The vast majority of these sites were located in tRNAs and a small number occurred in other non-coding RNAs (Fig. 4f). In contrast to all sites detected in the epidermal cells, most of these NSUN2-methylated RNAs, including VTRNA1.1, showed higher methylation levels in the differentiated condition (Fig. 4f–h). The higher level of

methylation was consistent in all replicates (Fig. 4g, h; right hand panels).

One explanation why specifically these NSUN2-dependent methylation sites were more abundant in differentiated cells was that they all occurred in abundant and stable non-coding RNAs. Due to their long half-life, the methylated forms might

**Fig. 4** VTRNA1.1 methylation and processing are altered during cell differentiation. **a** Treatment regime of keratinocytes using calcium switch assay. **b**, **c** qRT-PCR to measure RNA levels of up-regulated (**b**) and down-regulated (**c**) markers at 2 and 6 days after calcium treatment compared to the 0 day control. Error bars indicate s.d. ($n = 3$ qRT-PCRs) ****$p < 0.0001$, ***$p < 0.001$, **$p < 0.01$ multiple $t$-tests. **d** Methylation levels ($n = 5$ BS conversion reactions) at cytosines in RNA isolated from undifferentiated (undiff) and differentiated (diff) primary HK shown as box plots showing all points with minimum to maximum values. ****$p < 0.0001$ Mann Whitney test. **e** Log$_2$ coverage ($n = 5$ BS conversion reactions) of sites in RNA isolated from undifferentiated (undiff) and differentiated (diff) primary HK. **f** Correlation between methylation levels at cytosines in undifferentiated and differentiated primary HK. Elevated methylation levels at tRNAs (examples in blue) and VTRNA1.1 (red). **g**, **h** Methylation levels in VTRNA1.1 (**g**) and tRNA Leu CAA (**h**) in the indicated cells (left hand panels) and heat maps (right hand panels) showing methylation levels in the individual replicates. **i** Light microscope image comparing the morphology of primary HK transfected with a control siRNA (Ctr) or svRNA4 after 4 days of differentiation in high CaCl$_2$. Scale bar: 50 μm. **j** Western blot detecting KRT10 and OVOL1 in HK transfected with Ctr siRNA or svRNA4 four days after calcium-induction. Tubulin served as loading control. **k**, **l** Treatment regimes and transfection (upper panels) of svRNA4 (**k**) or anti-svRNA4 (**l**) and qRT-PCR (lower panels) to measure RNA levels of the indicated markers 4 days after calcium treatment. Error bars indicate s.d. ($n = 3$ qRT-PCRs). ****$p < 0.0001$, ***$p < 0.001$, **$p < 0.01$, *$p < 0.05$ student's $t$-test. Source data are provided as a Source Data file

accumulate during differentiation, provided the processing machinery 'reading' the presence or absence of the methyl mark was less active in differentiating cells. This hypothesis was supported by our finding that full-length VTRNA1.1 was more abundant in differentiated epidermal cells, while svRNA4 decreased (Fig. 4b, c). In addition, VTRNA1.1 was consistently better covered in differentiated cells by BS sequencing (Supplementary Fig. 4a). Similarly, the coverage of tRNA Leu$^{CAA}$ was reduced in undifferentiated cells and it also showed higher methylation levels at NSUN2-specific sites in the differentiated cell state (Fig. 4h; Supplementary Fig. 4b). Moreover, higher coverage also negatively correlated with expression of NSUN2 in the human dermal fibroblasts (Supplementary Fig. 4a, b). Thus, our data indicated that the overall methylation levels of stable non-coding RNAs depended on the presence of both the methylating enzyme and the RNA processing machinery.

**svRNA4 maintains undifferentiated transcriptional programme**. To provide direct evidence that both the presence of NSUN2 and VTRNA1.1 processing was required in the undifferentiated progenitor cells, we transfected a svRNA4 mimic into primary HK and differentiated them using the calcium switch assay (Fig. 4i). svRNA4-transduced keratinocytes failed to undergo the morphological changes that are normally associated with a stratified squamous epithelium (Fig. 4i)[43]. To confirm a reduced capacity to differentiate in the presence of svRNA4, we measured RNA and protein expression levels of terminal differentiation markers. Western blot for the differentiation marker KRT10 and OVOL1 revealed repression of both markers in the presence of svRNA4 (Fig. 4j). QRT-PCR further confirmed down-regulation of RNA expression levels of the terminal differentiation markers *Krt10*, *Inv*, and *Tgm* as well as *Nsun2* and *Myc*, both known to promote lineage commitment in skin (Fig. 4k)[9,44–46].

Importantly, inhibition of endogenous svRNA4 using an anti-svRNA construct was sufficient to reverse the abundance of the terminal differentiation markers of *Tgm*, *Inv*, and *Ovol1* even in low calcium growth conditions (Fig. 4l). The low abundance of full-length VTRNA1.1 in both experiments confirmed that both constructs efficiently repressed their target RNA sequences (Fig. 4k, l). Together, our data indicated that the presence of svRNA4 was sufficient to maintain the transcriptional programme of a committed, yet undifferentiated progenitor state.

**SRSF2 maintains cell cycling of progenitor cells**. Since the presence of svRNA4 modulated human epidermal differentiation and SRSF2 influenced svRNA4 processing, we next determined the functional role of SRSF2 during epidermal cell differentiation. We transfected the epidermal cells with SRSF2 siRNAs, and then induced them to differentiate by increasing the calcium concentration in the growth medium (Fig. 5a). As a control, we also

transfected a SRSF1 siRNA, and confirmed that only SRSF2 expression levels were reduced after knock-down with the SRSF2-specific siRNA on both RNA and protein levels (Fig. 5b; Supplementary Fig. 5a). As expected, the human keratinocytes underwent efficient differentiation and only *Ovol1* RNA levels were significantly increased in the absence of SRSF2 (Fig. 5c). These data indicated that the differentiation programme was largely unaffected by depletion of SRSF2.

Terminal differentiation of epidermal cells is defined by both the up-regulation of differentiation markers and exit from the cell cycle[47]. We noted that the SRSF2-depleted HK showed differences in the distribution of cell cycle phases when compared to the differentiated control keratinocytes (Supplementary Fig. 5b). Therefore, we next asked whether SRSF2 influenced cell division and removed *Srsf2* in undifferentiated HK for 48 and 72 h (Supplementary Fig. 5c). Only after 72 h of knock-down, we measured a significant up-regulation of the terminal differentiation markers *Krt10* and *Tgm* (Fig. 5d). In contrast, we found a consistent down-regulation of major cell cycle regulators as early as 48 h after transfection (Fig. 5e)[48,49]. Cell cycle analyses revealed that repression of SRSF2 led to a significant increase of the Sub-G1 phase of the cell cycle, indicating enhanced cell death of the transfected HK (Fig. 5f). Indeed, keratinocytes failed to survive longer than 72 h after SRSF2 depletion (Fig. 5g). Thus, SRSF2 is required for cell division and survival of primary human keratinocytes.

**NSUN2 and SRSF2 act in concert to process VTRNA1.1**. Finally, we asked whether the production of svRNA4 required both NSUN2 and SRSF2 in the primary human keratinocytes. Since depletion of SRSF2 caused substantial cell death after 72 h, we removed NSUN2 and SRSF2 for only 24 h to obtain the maximum number of viable cells (Fig. 5g). As described for the human dermal fibroblasts (hDF) (Fig. 1g), knock-down of NSUN2 in keratinocytes reduced the processing of VTRNA1.1 into svRNA4. However, the svRNA4 level was restored to normal, when we simultaneously repressed NSUN2 and SRSF2 for 24 h (Fig. 5h). The data was in line with our observation in human dermal fibroblasts, where removal of SRSF2 rescued the levels of svRNA4 in *NSUN2*−/− patient-derived cells (Fig. 3e). and confirmed that binding of SRSF2 to un-methylated VTRNA1.1 reduced its processing into svRNA4. We concluded that VTRNA1.1 processing required both NSUN2 and SRSF2 to maintain an undifferentiated cell state.

In summary, our study revealed that SRSF2 bound un-methylated VTRNA1.1 with higher affinity and thereby protected it from both methylation and cleavage. High levels of svRNA4 in the presence of both proteins allowed cell cycle progression but reduced expression of the differentiation-promoting transcription factor OVOL1 and other terminal differentiation markers (Fig. 5i).

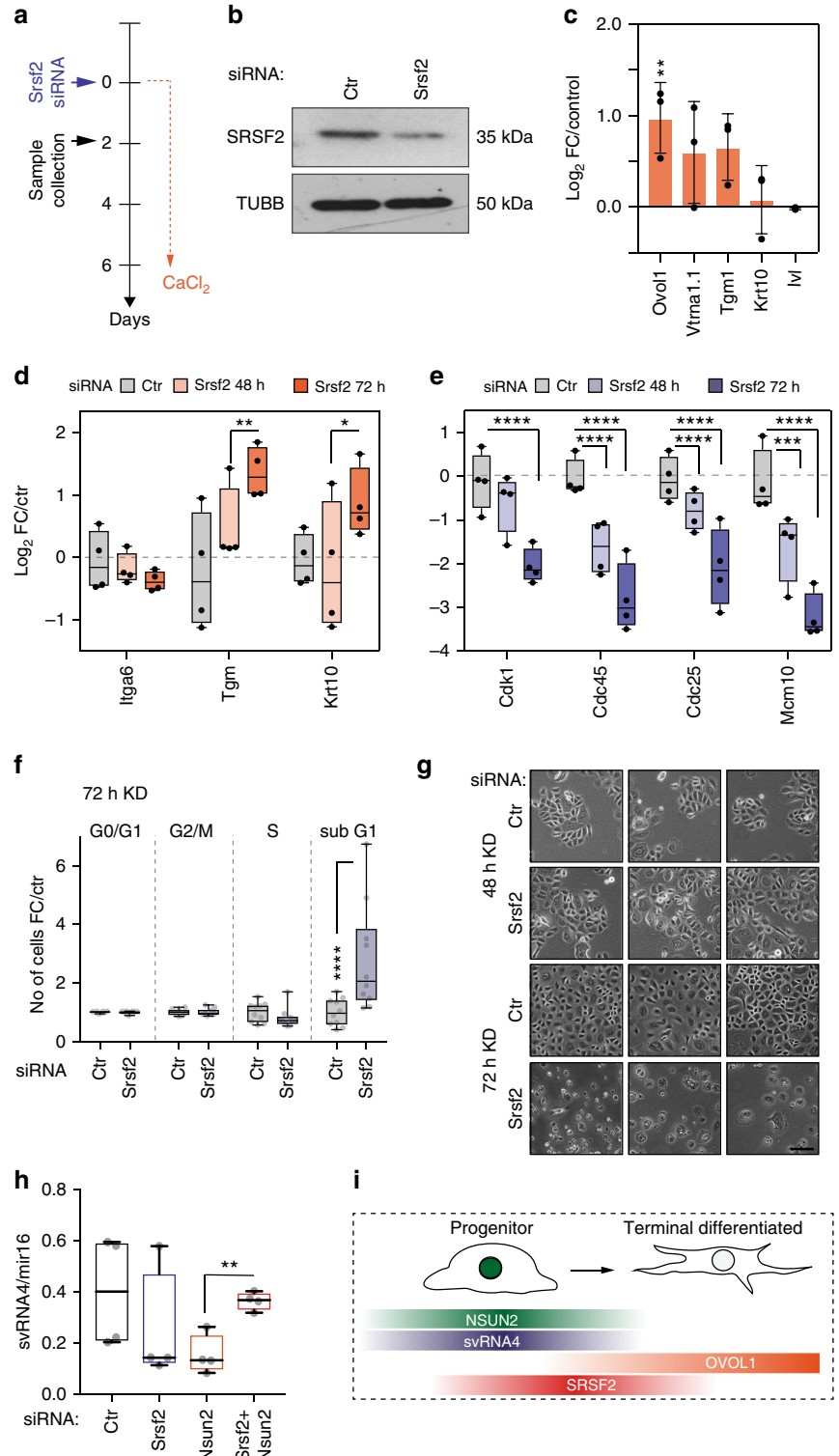

**Fig. 5** SRSF2 is required for cell cycle and survival of undifferentiated cells. **a** Treatment regime and transfection of differentiating primary human keratinocytes. **b** Western blot detecting SRSF2 after treatment with *Srsf2* siRNA. Tubulin (TUBB) serves as loading control. **c** Quantification of RNA expression levels of *Ovol1, Vtrna1.1, Tgm, Krt10,* and *Ivl* after 6 days of calcium-induced differentiation vs. untreated control (0 days). FC: Fold-change. Error bars indicate s.d. ($n = 3$ qRT-PCRs). **$p < 0.01$ two-way ANOVA. **d, e** Log$_2$ RNA fold-change (FC) of differentiation markers (**d**) and cell cycle regulators (**e**) after 48 and 72 h of *Srsf2* knock-down. Data shown as box plot with mean showing all data from minimum to maximum ($n = 4$). ****$p < 0.0001$, ***$p < 0.001$, **$p < 0.01$, *$p < 0.05$ Two-way ANOVA. **f** Cell cycle distribution of primary HK transfected with the *Srsf2* siRNA for 72 h. Error bars Data shown as box plot showing all data from minimum to maximum ($n = 10$ Flow sorts). ****$p < 0.0001$ Two-way ANOVA. **g** Light microscope images of HK transfected with a control (ctr) siRNA (upper panels) and a *Srsf2* siRNA (lower panels) after 48 and 72 h in low calcium medium. Scale bar: 50 µm. **h** Small RNA qRT-PCR measuring the abundance of svRNA4 in primary HK transfected with the indicated siRNA constructs. Data shown as box plot with mean showing all data from minimum to maximum. Error bars represent s.d. ($n = 4$ qRT-PCRs). **$p < 0.01$ unpaired student's *t*-test. **i** Illustration how levels of NSUN2, svRNA4, SRSF2, and OVOL1 change upon terminal differentiation in keratinocytes. Source data are provided as a Source Data file

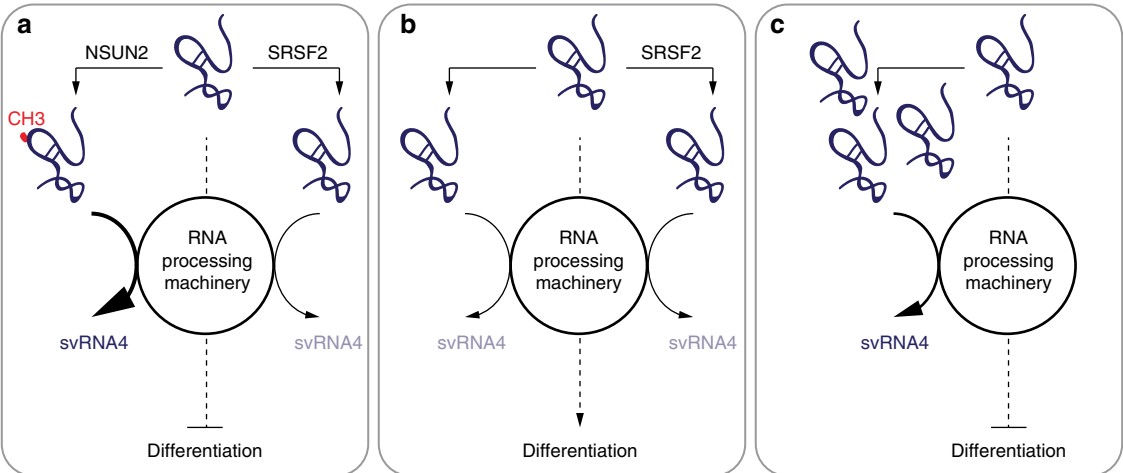

**Fig. 6** Summary of VTRNA1.1 processing into svRNA4. **a** Expression of both NSUN2 and SRSF2 (e.g. in progenitor cells) result in high levels of VTRNA1.1 methylation ($CH_3$) and high levels of svRNA4 and repressed differentiation. **b** No NSUN2 in the presence of SRSF2 suppresses formation of svRNA4 and allows differentiation. **c** Lack of expression of both NSUN2 and SRSF2 release VTRNA1.1 from SRSF2 binding and increases the levels of svRNA4

Thus, the presence and expression levels of both NSUN2 and SRSF2 coordinated how much methylated VTRNA1.1 was processed into svRNA4 (Fig. 6a–c).

## Discussion

The proper formation of $m^5C$ in tRNA is required for normal development[7]. Aberrant deposition of $m^5C$ into tRNAs causes neuro-developmental deficits by impairing the translation machinery[2,8]. While the functional role of $m^5C$ in tRNA is now increasingly understood, the importance of $m^5C$ in other non-coding RNAs remains unclear. NSUN2 is one of the best char-acterised cytosine-5 RNA methylases and methylates the vast majority of tRNAs. Here, we confirm that NSUN2 also methylates the vault RNA VTRNA1.1 in a wide range of human cells. NSUN2 is the sole enzyme to methylate cytosine 69 in VTRNA1.1 and thereby regulates its processing into multiple regulatory small RNAs (svRNAs).

RNA modifications control the fate and function of RNA molecules by recruiting or repelling specific RNA binding pro-teins[50]. To determine the functional relevance of $m^5C69$ on VTRNA1.1 metabolism, we identified all RNA-binding proteins that bind methylated or un-methylated VTRNA1.1 with different affinity. Our quantitative mass spectrometry-based approach identified SRSF2 as a novel VTRNA1.1 binding protein that was repelled by the presence of $m^5C69$. During gene transcription, SRSF2 contributes to constitutive and alternative splicing by binding to exonic splicing enhancer sequences (ESE), pre-dominantly within intron-containing pre-mRNA[33,34]. In addi-tion, SRSF2 binds mRNAs via non-ESE sites, for instance the HIV-1 tat mRNA[51]. We identified two SRSF2 binding sites in VTRNA1.1, one of which overlapped with the methylated C69. SRSF2 has been described to bind other non-coding RNAs such as 7SK and MALAT1[30–32]. The binding of MALAT1 or 7SK is thought to influence the recruitment of SRSF2 to distinct active transcriptional regions[30,32]. Nuclear un-methylated VTRNA1.1 might have a similar function.

In addition to their roles in splicing, SR proteins are thought to have a broader role in RNA metabolism. SRSF1, for instance, facilitates the processing of certain microRNAs independent of its function in splicing[52]. Furthermore, CLIP assays revealed little direct correlation between SRSF2 protein binding and induced splicing changes[31]. Here, we propose that SRSF2 contributes to VTRNA1.1 processing into svRNA4 binding to un-methylated VTRNA1.1 and thereby protecting it from methylation by NSUN2 (Fig. 6a). For instance, epidermal progenitors express high levels of both NSUN2 and SRSF2 and consequently, methylate a substantial fraction of VTRNA1.1. Methylated VTRNA1.1 binds SRSF2 with lower affinity and the levels of svRNA4 increase. Deletion of NSUN2 leads to loss of $m^5C$ at C69 and enhances the binding of VTRNA1.1 to SRSF2 (Fig. 6b). Binding to SRSF2 reduced the level of VTRNA1.1 entering the RNA processing machinery leading to reduced production of svRNA4. Finally, we tested a scenario where both proteins are repressed (Fig. 6c). This enhanced the fraction of free VTRNA1.1 that entered the RNA processing machinery leading to increased levels of svRNA4. Together, VTRNA1.1 methylation at C69 facilitates the production of svRNA4 by reducing SRSF2 affinity to VTRNA1.1.

Deciphering the precise function of SRSF2 in modulating the processing of VTRNA1.1 was hampered by the fact that SRSF2 is an essential protein upstream of both RNA methylation and processing. Homozygous germ line deletion of SRSF2 is embryonic lethal[53], and conditional knock-out mice display tissue-specific phenotypes[53–56]. Similar to repression of SRSF2 in epidermal progenitors, loss of SRSF2 in mouse embryonic fibroblasts induced G2/M cell cycle arrest and genomic instabil-ity[54]. In addition, SRSF2 is required to maintain human embryonic stem cell pluripotency[57]. Mutations in the human *SRSF2* gene altering its RNA-binding affinity impairs hemato-poietic differentiation in vivo and is frequently (40% incidence) found in patients with myelodysplastic syndromes and certain leukaemias[28,58,59].

The function of SRSF2 in splicing might be modulated by VTRNA1.1, yet VTRNA1.1-methylation and processing are probably not linked to splicing. Repression of SRSF2 in primary human keratinocytes caused cell cycle arrest and cell death, and this effect was independent of VTRNA1.1-methylation and processing. However, simultaneous deletion of SRSF2 and NSUN2 increased svRNA4 levels confirming that SRSF2 is involved in regulating VTRNA1.1 processing in primary human keratinocytes. Thus, NSUN2 and SRSF2 together determined how much VTRNA1.1 was processed into svRNAs. Finally, we confirmed a functional relevance of svRNA4 in regulating epi-dermal differentiation. One explanation for how svRNA4 modulated epidermal differentiation is through post-transcriptional silencing of mRNA targets. Our previous bioin-formatic analysis identified OVOL1 as a high confidence target of

svRNA4[3]. While we indeed found an inverse relationship between the abundances of OVOL1 and svRNA4, more work is needed to confirm that svRNA4 indeed acts similarly to miR-NAs. However, we identified a functional relevance for svRNA4 in regulating the terminal differentiation program. While enhanced levels of svRNA4 repressed the terminal differentiation program, sequestering svRNA4 was sufficient to trigger differentiation.

## Methods

**Cell culture, transfections and infections**. HEK293 and HeLa cells (ATCC) were grown in Dulbecco's Minimal Essential Medium (DMEM) (Thermo Fisher Scientific) supplemented with 10% foetal bovine serum (FBS) (Sigma), penicillin, and streptomycin. Human dermal fibroblasts[11] were grown in Minimal Essential Medium (MEM) (Thermo Fisher Scientific) supplemented with 20% FBS (Sigma), 1% penicillin, and streptomycin. The human embryonic stem cell line Hues9 (H9) was obtained from the WiCell. H9 cells were maintained in Essential 8 media (Thermo Fisher Scientific) on human embryonic stem cell-qualified matrigel (Corning) coated plates at 37 °C, 5% $CO_2$. Primary human keratinocytes isolated from neonatal foreskins (Cellworks distributed, ZHC-1116) were cultured on collagen (BD Biosciences) coated plates in KGM-gold (Lonza) medium. For cell propagation calcium concentrations were adjusted to 0.06 mM using 1.2 M $CaCl_2$ stock solution. Medium was changed every other day and cells were passaged when reaching 60–70% confluency. For epidermal differentiation, KGM-gold calcium concentrations were adjusted to 1.2 mM using 1.2 M $CaCl_2$ stock solution. Primary human epidermal cells were only used until passages 6–7 after thawing. All cells were cultured in humidified atmospheres with 5% $CO_2$.

For the transfection of small interfering RNA (siRNA), human dermal fibroblasts and primary human keratinocytes (HK) cells were cultured until reaching 50–60% confluency. Cells were then transfected with a control siRNAs or svRNA4 antagomirs (2′-O-methyl-anti-svRNA4: 5′ AAA AGG ACU GGA GAG CGC CCG CGG GUC UCG); control microRNA miRIDIAN mimic or svRNA4 microRNA miRIDIAN mimic (5′ CGA GAC CCG CGG GCG CUC UCC AGU CCU UUU) (Dharmacon-GE); SRSF1 siRNAs (QIAGEN) or SFSF2 siRNAs (QIAGEN, and Dharmacon-GE) using RNAimax transfection kit (Thermo Fisher Scientific). 24, 48, or 72 h post-transfection, cells were washed in PBS and RNA or protein was isolated.

For the infection of the short hairpin (sh) RNA, the most effect siRNAs were clones into PLKO.1 puro plasmids (Addgene). To produce the lentivirus, HEK293T cells were grown in DMEM supplemented with 10% FBS on 60 cm cell dishes until 40–50% confluent and then transfected with 15 µg lentiviral PLKO.1 puro vectors containing shRNA for SRSF1 (shRNA1.1, shRNA1.6) or SRSF2 (shRNA2.4, shRNA2.5) or the empty PLKO.1 puro vector and 7.5 µg of both packaging vectors (Pol and Gag) using the calcium-phosphate transfection kit (Thermo Fisher Scientific). The next day the medium was carefully changed to avoid detaching HeK293T cells and kept in culture for another 24 h. Subsequently, supernatant containing viral particles were collected and filtered (0.22 µm) before being used to infect human dermal fibroblast. The next day, the medium was changed, and cells were kept in culture for 24 h before selecting with 1 µg/ml puromycin for 4 days with medium changed after day 2. After selection, the fibroblasts were propagated without purpmycin for further experiments.

To rescue expression of NSUN2 protein or its enzymatic dead version, full-length human NSUN2 (pB-NSUN2), inactive mutants C271A (pB-NSUN2-C271A), K190M (pB-NSUN2-K190M) or C321A (pB-NSUN2-C321A), and the empty vector (pB-empty) were infected retrovirally[26].

**RNA bisulfite sequencing**. All RNA bisulfite conversion experiments were performed in five independent replicates. Total RNA of about 4 µg was extracted using Trizol (Thermo Fisher Scientific) and then DNase (Ambion) and Ribo-Zero (Illumina) treated according to the manufacturers' instruction. The remaining RNA fraction was bisulfite-converted. Briefly, ribosomal depleted RNA was mixed with 70 µl of 40% sodium bisulfite pH 5.0 and DNA protection buffer (EpiTect Bisulfite Kit, Qiagen). The reaction mixture was incubated for three cycles of 5 min at 70 °C followed by 1 h at 60 °C and then desalted with Micro Bio-spin 6 chromatography columns (Bio-Rad). RNA was desulphonated by adding an equal volume of 1 M Tris (pH 9.0) to the reaction mixture and incubated for 1 h at 37 °C, followed by ethanol precipitation. Bisulfite converted RNA was then treated with T4 PNK (New England Biolabs) to repair both 5′ and 3′ ends for library preparation. Repaired RNA quality and concentration was measured on a Bioanalyzer 2100 RNA nano-chip (Agilent). About 100 ng of RNA was used to generate the libraries using the TruSeq Small RNA preparation kit (Illumina). RNA adapters were then ligated, reverse-transcribed and amplified by 18 cycles of PCR before sequencing on a HiSeq4000 (Illumina).

**Small qRT-PCR and qRT-PCR**. To measure the abundance of svRNA1 and svRNA4 we performed small quantitative PCR (qRT-PCR). Total RNA from human epidermal cells, NSUN2+/−, NSUN2−/− dermal fibroblasts was isolated using TRIzol according to the manufacturer's instruction (Thermo Fisher

Scientific). RNAs of smaller than 200 base pairs were enriched using mirVana kit (Ambion) according to the manufacturer's instructions. Enriched RNAs were then separated using 15% acrylamide-urea gels (Thermo Fisher Scientific) and gel slices spanning 10–40 bps were excised and crushed in RNase free water and incubated overnight at 4 °C. Eluted RNA was then purified using spin-X centrifuge tube filters (0.22 µM) (Costar) and ethanol precipitated over night at −20 °C. Small RNA reverse transcription and qRT-PCR was performed as follows[60]. PolyA tails were added to small RNAs using poly(A) polymerase (Ambion). The poly(T) primer 5′-GCG AGC ACA GAA TTA ATA CGA CTC ACT ATA GG(T)12VN-3′ was then used for cDNA synthesis. QuantifastSYBR® green master mix (QIAGEN) was used for qRT-PCR reactions. cDNA was incubated at 90 °C for 10 min followed by 40 cycles of 30 s denaturation at 90 °C, and 1 min of extension at 60 °C. The forward primers were: miR-16 (5′-TAG CAG CAC GTA AAT ATT GGC G-′3), svRNA1 (5′-TGT CTG GGT TGT TCG AGA CCC GCG GGC-3′), and svRNA4 (5′-CGA GAC CCG CGG GCG CTC TCC AGT CCT TTT-′3). The reverse primer for all reactions was: 5′-GCG AGC ACA GAA TTA ATA CGA C-3′.

For conventional quantitative PCR (qRT-PCR) total RNA was isolated from human fibroblasts, and primary human keratinocytes cells using TRIzol (Thermo Fisher Scientific). cDNA synthesis was performed using the SuperscriptIII reverse transcriptase kit (Thermo Fisher Scientific) according to manufacturer's instructions. qRT-PCR was performed using TaqMan assay sets purchased from Thermo Fisher Scientific and used as per manufacturer's recommendations or pre-designed primers (Sigma-Aldrich) and Sybr master mix 2 × (Life Technologies) were used. The following probes were used to amplify selected genes: Gapdh (4326317E), Nsun2 (Hs00214829_m1), Srsf2 (Hs00958207_cn), Srsf1 (Hs00199471_m1), Ovol1 (Hs00190060_m1), vt-RNA1.1 (Hs03676993_s1), Inv (Hs00846307_s1), Tgm1 (Hs00165929_m1 and Hs01070316_m1), Krt10 (Hs01051614_g1 and Hs00166289_m1), Myc (Hs00153408_m1), Itgα6 (Hs01041013_m1), Cdc45 (Hs00907337_m1), Mcm10 (Hs00960349_m1), Cdc25 (Hs00156411_m1), and Cdk1 (Hs00938777_m1). The following pre-designed primers were used: Gapdh (forward: ATC TTC CAG GAG CGA GAT CC, reverse: ACC ACT GAC ACG TTG GCA GT), Srsf2 (forward: CCT AAT TTG TGG CCT CCT GA, reverse: TCA ATC TCT TGA CAG CT TAG GC).

**SILAC**. HeLa cells were grown in SILAC DMEM (Thermo Fisher Scientific) supplemented with either "light" L-lysine-2HCl and L-arginine-HCl or "heavy" 13C6-L-lysine-2HCl and 13C615N4-L-arginine-HCl. For full incorporation of "light" or "heavy" L-Lys/L-Arg, for at least six passages. Subsequently, $3 \times 10^6$ cells were resuspended in 1 ml of buffer-D (100 mM Tris-HCl at pH 8.0; 100 mM KCl; 0.2 mM EDTA; 0.5 mM DTT; 0.2 mM PMSF; 20% (w/v) glycerol), scraped and sonicated (Diagenode). The suspension was centrifuged for 5 min at 10,000 × g, and the supernatant was used for VTRNA pull-down assays followed by mass spectrometry detection[61]

**RNA pull-down and SILAC mass-spectrometry (RP-SMS)**. We performed agarose mediated VT-RNA1.1 pull-downs[62,27]. Briefly, 10 µg of methylated, or un-methylated VT-RNA1.1 (Dharmacon-GE) was treated with sodium m-periodate (Sigma) for 1 h at room temperature while rotating. VT-RNAs were then precipitated by 3 M of sodium acetate (Ambion) and ethanol. Recovered VT-RNAs were then covalently coupled with adipic acid dihydrazide agarose beads (Sigma) overnight at 4 °C. VT-RNA-bead complex was incubated with 40% v/v of total SILAC "heavy" HeLa lysates supplemented with 1.5 mM $MgCl_2$, 25 mM creatine-phosphate (Millipore), 200 units per ml RNAseOUT (ThermoFisher Scientific) and 5 mM ATP (Sigma) for 30 min at 37 °C. The beads only control was incubated with SILAC "light" HeLa lysates. The reactions were washed four times with Buffer-G (20 mM Tris-HCl at PH7.5; 137 mM NaCl; 1 mM EDTA; 1% Triton x100; 10% glycerol; 1.5 mM $MgCl_2$) or Buffer-A (20 mM Tris-HCl at pH: 7.5; 50 mM NaCl; 1 mM EDTA; 1% NP-40; 10% glycerol; 1.5 mM $MgCl_2$), for less stringent RNA-protein interactions. After the final wash, RNA-bead complex was mixed with beads only control and incubated in 60 µl $H_2O$ containing 4x LDS protein sample buffer (Novex), 10X reducing reagent buffer (Novex), then incubated at 70 °C for 10 min. The end solution was centrifuged at maximum speed for 1 min and the supernatant was collected and analysed either by quantitative mass-spectrometer or SDS-PAGE western blot for proteins detection[27].

**RNA immunoprecipitation (RIP)**. Pelleted human dermal fibroblasts were lysed in RIP buffer (1% NP-40; 25 mM Tris-HCl at pH 8.0; 150 mM NaCl; 2 mM MgCl2; 1 mM DTT) and incubated on ice for 30 min. SRSF2 antibody (5µ ab11826; Abcam), or control serum were pre-incubated with Protein G magnetic dynabeads (Thermo Fisher Scientific) in lysis buffer for 30 min. Following washing of beads, cleared lysates were added to the beads and immunoprecipitation was carried out for 2 h at 4 °C with gentle mixing. Beads were then washed extensively in lysis buffer and precipitated RNAs were recovered from the beads using TRIzol (Thermo Fisher Scientific). Immunoprecipitation reactions were performed in the presence of 200 units per ml of RNAse inhibitor.

**Co-immunoprecipitations and western blot**. Confluent 150 cm dishes of cells were washed twice with ice-cold PBS before being scraped using 300 µl of ice-cold

CHAPS (FIVE photon Biochemicals) lysis buffer supplemented with protease inhibitors (Roche) and phosphatase inhibitors (Roche). Lysates were transferred to pre-chilled Eppendorf tubes and incubated on ice for 10 min. Tubes were strongly taped several times during the incubation period to facilitate cell membrane lysis. Lysates were centrifuged for 15 min at maximum speed under cool conditions. The supernatant was stored until used for co-IP analysis. About 3 μg of primary antibodies or control pre-bled serum were washed three times with PBS before incubated with CHAPS prepared HeLa lysates at 4 °C for 1 h. During incubation period 50 μl of Agarose Anti-Rabbit IgG IP beads (ROCKLAND) was washed extensively with pre-chilled lysis buffer (50 mM Tris-HCl pH: 8; 150 mM NaCl; 1% NP-40; protease inhibitor; phosphatase inhibitor). After the incubation, Agarose Anti-Rabbit IgG IP beads were added to the mixture and left rotating overnight at 4 °C. The samples were then centrifuged at $1000 \times g$ for 1 min and the supernatant was removed. The pelleted bead-antibody complex was washed four times with lysis buffer. The bead-antibody complex was then mixed with 4× LDS sample buffer (Novex), and 10× reducing reagent buffer (Novex) and incubated at 70 °C for 10 min for western blotting detection.

For western blotting, cells were harvested and washed twice with PBS before being lysed with lysis buffer (50 mM Tris-HCL at pH 7.4; 250 mM NaCl; 1% NP-40; 0.1% SDS; 0.5% sodium deoxycholate) and were cleared by centrifugation at full speed for 10 min at 4 °C. Supernatant was mixed with 4× LDS protein sample buffer (Novex), 10× reducing reagent buffer (Novex) and incubated at 70 °C for 10 min. Samples were run on running buffer (Novex) containing 40× NuPAGE Antioxidant (Novex) then electrophoresed on a 4–12% Bis-Tris precast polyacrylamide gels (Novex). Proteins were transferred onto nitrocellulose membranes (GE Healthcare) using transfer buffer (191 mM Glycine; 25 mM Trishydrochloride; 10% Methanol; 0.1% SDS). Nitrocellulose membranes were blocked in 10% western Blocking Reagent (Roche). Blots were then incubated with primary antibodies in blocking solution overnight at 4 °C then followed by incubation with the appropriate HRP-conjugated secondary antibodies. The chemiluminescent signal was detected using an ECL chemiluminescent kit (GE Healthcare) according to instructions. The following antibodies were used: NSUN2 (1:1000; Met-A)[63], hnRNP-A1 (1:2000; D21H11; Cell Signalling Technology), SRSF2 (1:1000; ab11826; Abcam), SRSF1 (1:2000; 32–4500; Thermo Fisher Scientific), OVOL1 (1:1000; ab65023; Abcam), KRT10 (1:2000; PRB-159P; Covance), Tubulin (1:5000; clone DM1A; Sigma), PUS7 (1:2000; ab118039, Abcam).

**EMSA**. EMSA was performed with end-labelled vtRNA1.1 and indicated amounts of recombinant GST-SRSF2 RRM. Probes ($50 \times 10^3$ counts per minute, ~0.1 pmol) were incubated in 16 μl reactions with recombinant protein in elution buffer (25 mM Tris-HCl, 5 mM reduced glutathione, pH 8.0) supplemented with 3 mM $MgCl_2$, 0.5 mM ATP, 37.5 mM creatine phosphate and 10 ng tRNA for one hour on ice. Reactions were mixed with native loading buffer and analysed on a 6% (w/v) non-denaturing polyacrylamide gel run in 0.5×TBE at 8 W for 1 h 10 min. The signal was registered using radiographic X-ray film.

**Cell cycle analyses**. Flow cytometry was used to probe cell cycle stage. Flow cytometry analysis was performed with the LSRFortessa Flow Cytometer (BD Biosciences). Cells were washed in PBS and collected with Trypsin-EDTA (1:1 in PBS). Cells were then fixed by resuspending in ice-cold 70% ethanol. Samples were kept at 4 °C until processing. Before processing cells were centrifuged at $12,000 \times g$ for 5 min and resuspended in 3 mL PBS with DAPI (1:3000). Fluorescence of each samples was measured on the flow cytometer at 450/50 405 nm. Data was analysed using FCS express6 (DeNovo Software). All samples were gated using forward vs. side scatter to eliminate debris.

**Bioinformatic analyses**. To assess the transcriptome-wide methylation levels and to create the methylation level heatmaps, Trim Galore! (v0.4.0) with parameters "--stringency 3 -e 0.2 -a TGGAATTCTCGGGTGCCAAGGA" was used to remove sequencing adapters and reads shorter than 20 nucleotides. Alignment to the hg38 reference genome was done with Bismark (v0.14.4) with parameters "-n 2 -l 50 --un --ambiguous --bowtie1 --chunkmbs 2048", to allow for up to two mismatches and to save unaligned and ambiguously mapping reads separately. Seqtk with parameters "-e 3" was used to remove the last three bases (a potential 'CCA' tail) from the unaligned and ambiguous reads followed by a second alignment attempt using Bismark. Finally, ngsutils (v0.5.9) in the "junction" mode was used to extract splice junctions from known genes (Gencode v28) and unaligned reads from the second attempt were aligned to the junctions using Bismark. The aligned reads were converted back to genomic coordinates using bamutils in "convertregion" mode. 'N' in the cigar string was replaced with 'D' for compatibility with bismark_methylation_extractor.

To identify novel high-confidence methylation sites we selected all sites with read coverage above 10 in all replicates. Welch's $t$-test was used to compare the methylation levels of either the empty vector (e. Vector) or the enzymatic dead version (K190M) to the Nsun2 rescue (NSUN2); five replicates per group. False discovery rate was used to correct for multiple testing. Methylation sites with padj $< 0.05$ were considered to be differently methylated.

Samtools merge was used to combine aligned reads from all three steps. Reads with >1/3 methylated cytosines were discarded as they are likely artifacts from highly

structured regions. The bismark_methylation_extractor with the "--bedGraph --counts --CX_context" options was used to extract methylated cytosines.

Sequencing data from patient cells are deposited on dbGaP under the accession number phs000645.v4.p1. The sequencing data are available on GEO with the accession numbers GSE122600 and GSE125046.

Gene set enrichment analyses were done using GOrilla [http://cbl-gorilla.cs.technion.ac.il/][64]. As running mode, we used "two ranked lists of genes" using all identified binding proteins as background list.

**Reporting summary**. Further information on research design is available in the Nature Research Reporting Summary linked to this article.

## Data availability
Patient-related sequencing data are available on dbGaP: phs000645.v4.p1. All other sequencing data are available on GEO: GSE122600 and GSE125046. The source data underlying Figs. 1c, g, h; 2c-f; 3a-e; 4b, c, j-l; 5b-f, h and Supplementary Figs. 1a; 2d, f; 3a-c; 5a and c are provided as a Source Data file. All data is available from the authors upon reasonable request.

## Code availability
The scripts used for the alignment and processing of the bisulfite sequencing data are available at [https://github.com/susbo/trans-bsseq].

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

## Acknowledgements

We thank everybody who provided us with reagents, in particular we thank James Stevenin for sending us recombinant SRSF2. We gratefully acknowledge the support of all the Wellcome - MRC Cambridge Stem Cell Institute core facility managers. This work was funded by Cancer Research UK (CR-UK) and the European Research Council (ERC). Parts of this research in Michaela Frye's laboratory was supported by core funding from Wellcome and MRC to the Wellcome-MRC Cambridge Stem Cell Institute. Juri Rappsilber's laboratory was supported by a Wellcome Trust Senior Research Fellowship (084229). Gracjan Michlewski's laboratory was supported by the MRC Career Development Award (G10000564), Wellcome Trust Seed Award (210144/Z/18/Z) and Wellcome Trust Centre for Cell Biology Core Grants (077707 and 092076). A.S. was supported by a scholarship from the University of Tabuk and Khalifa University of Science and Technology Faculty start-up award number FSU-2018-01. R.E.W. was supported by the Wellcome Trust PhD Programme in Stem Cell Biology & Medicine.

## Author contributions

M.F. and G.M. designed and analysed data and wrote manuscript. A.A.S. performed the majority of the experiments and analysed data. N.R.C., R.E.W. and T.S. performed experiments and analysed data. S.B. and S.D. performed computational analyses. C.S. and J.R. performed and supervised analyses.

## Additional information

**Competing interests:** M.F. consults for STORM Therapeutics. The remaining authors declare no competing interests.

