## [Peer Review File · Nature Communications]

Reviewers' comments:

Reviewer #1 (Remarks to the Author):

Post-transcriptional RNA modifications have gained recent limelight because technological advances allow measuring and even mapping various RNA modifications in their sequence context. While most of the recent work in the RNA modification field has been performed on elucidating the positional identity of methyl-6 adenosine (m6A), another RNA modification, 5-methylcytosine (m5C), is only been studied by a few research groups.

While m5C in tRNAs and rRNAs is an accepted modification of these abundant RNAs, only a handful of papers reported on attempts to mapping m5C in other RNAs, mostly by transcriptome-wide approaches. In these data sets, tRNAs and rRNAs were confirmed to contain m5C, however, results regarding the identification of m5C in other and less abundant RNAs were sobering and mostly non-overlapping. This indicated that the placement of m5C in non-abundant RNAs is either non-existing, or too low in quantity to be robustly detected or displays a rather dynamic, tissue-or cell type-specific or, growth condition or stimuli-dependent distribution. The biological function for m5C, even in tRNA or rRNA remains rather unclear. A common denominating principle that was derived from work on tRNA methylation suggests that m5C contributes to the stability of certain tRNAs, especially during stress responses indicating that RNA processing enzymes can read m5C.

NSun2 is a (cytosine-5) methyltransferase that acts mostly on tRNAs. A few alternative substrates, most of them non-coding RNAs, have been identified using transcriptome-wide approaches such as miCLIP (the authors of the presented manuscript), Aza-IP and RNA bisulfite sequencing. This referee is aware of a manuscript by Xin Yang et al., Pubmed: 28418038, which recently reported on robust NSun2-dependent messenger RNA methylation in HeLa cells. However, since it was also reported that many cancer cell lines (including HeLa cells) show genomic gains of the NSun2 locus, it stands to reason that a number of the identified m5C sites in these cancer cell lines are not physiological but are in fact artefacts caused by RNA methyltransferase overexpression.

This notion has to be taken also into account when performing experiments in HEK293 cells, the cell system used for the identification of vault RNA (vtRNA) methylation by NSun2 as reported by Hussain et al., 2013. In this paper, two out of four vtRNAs contained m5C as confirmed by targeted RNA bisulfite sequencing. Interestingly, loss of NSun2-mediated RNA methylation on only one of these vtRNAs (vtRNAs 1.1) affected its processing into smaller RNAs (svRNAs). And rather counter-intuitively, and contrary to the one-sided effects of NSun2-mediated RNA methylation on tRNAs, loss of NSun2 both increased and decreased the stability of vtRNAs1.1 as measured by the abundance of 4 different small RNAs that were all derived from vtRNAs1.1.

The manuscript by Sajini et al. is now building on these findings and asks how does NSun2-mediated methylation on vtRNAs1.1 affect its processing into one particular svRNA (svRNA4) and what would be the biological function of svRNA4?

The work is important because very little is known about the exact function of particular RNA modification sites. Furthermore, it is of utmost importance to begin analysing the proteins that mediate the function of modified RNAs. In the presented work, the authors identify the factor SRSF2 as vtRNA1.1 binding protein, which cooperates in processing of vtRNAs1.1.

The authors claim that NSun2-mediated RNA methylation fine-tunes binding of this factor to vtRNAs1.1 thereby influencing the extent of vtRNAs1.1 processing and the levels of svRNA4 during the process of epidermal differentiation.

While these findings would generally warrant publication in Nature Communications, the presentation of experimental evidence is presently insufficient. Some experiments contain technical caveats, which do not allow for the conclusions the authors make and therefore curtail the scientific value of this study.

Therefore, this referee concludes that the authors have to thoroughly revise the manuscript before

it can be (re)-considered for publication in Nature Communications.

General comments:

> The wording of the manuscript title indicates a seemingly wide and review-like scope of the work. The title should be changed to a more pointed statement because the manuscript actually does not address the writing and reading of 5mC in RNA but rather that a lack of 5mC on vtRNAs is determining its processing.

> The reasoning for the experimentation, especially after the initial identification of proteins binding to un-methylated versus methylated vtRNAs1.1 is rather convoluted and often hard to follow. This referee suggests restructuring into more than the existing 4 subheadings.

> Given the introduction to m5C, NSun2 and the potential biological function of the established RNA methylation circuits, it remains largely unclear to the reader why the authors chose to study a protein (SRSF2) that does un-methylated RNA rather than methylated RNA. However, only after looking at the data in Figure 1C and the Figure Legend the reader realises that the only proteins that were repeatedly identified in two independent RNA pull-down experiments (asterisk) on SILAC-treated cells followed by mass spectrometry were SRSF2 and PUS7. That all other proteins, which showed a greater differential in peptide counts than SRSF2, might have bound disadvantageously to vtRNA1.1 is a major concern for the interpretation of the robustness of the used initial approach, especially in light of the fact that SRSF2 seems to bind to both un-methylated and methylated vtRNAs1.1 (Figure 1D) and that the log₂ value of the fold-change (FC) of identified peptides is 0.5 for SRSF2 and 0.7 for PUS7 in the differential RNA pull-downs. These are indeed very small values and raise the question as to the reproducibility of these initial results, which, importantly, define all other experiments in the manuscript.

> VTRNA1.1 methylation was identified using HEK293 cells. However, the presented manuscript uses also HeLa cells, primary human keratinocytes and their differentiated progeny. Nowhere in the manuscript is a mention of the methylation status of VTRNA1.1 in these cell types, only the assumption that this RNA is methylated as observed in HEK293. The authors should provide (quantitative) RNA methylation data (preferably by RNA bisulfite sequencing) where needed (see below), especially in light of a "magic" factor of 2 that is defining the binding of SRSF1 to un-methylated/methylated VTRNA1.1. From the original publication by Hussain et al., 2013 this referee gathers from the heat map that the methylation levels might be at around between 25 to 50% at position C69. Could that partial methylation help the authors explain why SRSF1 is binding only a little bit better to the un-methylated RNA?

> Please, double-check all Figure calls. For instance, on page 5: Fig. 1B is wrongly called as Fig. 2B.

> Please, double-check all Reference calls. For instance, on page 12: citation (47) is not correct in the context of the statement of the sentence.

> RNA scientists like to see the RNAs they study usually on Northern blots. VTRNA1.1 can be detected nicely by northern blotting (Amort et al., 2015) and this referee wonders whether or not northern blotting would substantiate some of the claims the authors make (see below).

> Referees should have access to primary data, especially when major conclusions are based on such data. In the case of the presented manuscript, there is no indication what the read counts from the mass spec data are, which is important to judge certain Figures submitted with the manuscript.

Specific comments:

> Page 5: "The processing of VTRNA1.1 into svRNA4 depended on the methylation activity of NSUN2 because only the wild-type construct of NSUN2 increased svRNA4 production (Fig. 1B)."

>> To claim that methylation activity is involved the authors should provide RNA bisulfite data on VTRNA1.1 for each experiment to show that NSun2^{-/-} cells regain m5C when overexpressing wt

NSun2.

> Page 5: "Thus, the presence of a methylation group at C69 determined the processing of VTRNA1.1 into svRNA4."

>> It would be helpful to show Northern data for the 29 nt small svRNA4 to substantiate the claim that this small RNA is being produced in a methylation-dependent fashion.

> Page 6: "...we performed RNA pull-down assays, and confirmed that SRSF2 bound non-methylated (C69) with higher affinity than methylated (m5C69) VTRNA1.1 (Fig. 1D)."

>> Since the authors prepared the RNA bait themselves using activated agarose and have not provided data about the efficiency of RNA coupling it remains unclear whether similar molarities of methylated and un-methylated vtRNA1.1 have been subjected to the assay. Therefore, it is wrong to call the presented data measured affinities between RNA and protein because SRSF2 protein was probed by western blotting. If the authors would like to keep the wording "affinities" they need to present data containing experiments with equimolar SRSF2 protein concentrations on equimolar mass of bait RNA. If the authors cannot contribute such data, they need to tone down the scientific description of the data.

> Page 6: "hnRNP A1 protein served as a control as it displayed equal binding to m5C69 and C69 VTRNA1.1."

>> The authors should mention that hnRNPA1 binds to vtRNA1.1. Either by referring to previous published data or to their own data from the SILAC pull-down experiments.

> Page 6: "VTRNA1.1 contains two putative SFSR2 RNA binding motifs (RRM1 and RRM2), one of which overlapped with the methylated cytosine 69 (Fig. 1E)."

>> The authors should provide a reference as to how such SFSR2 binding motifs are defined.

> Page 6: "To validate the functional importance of the SRSF2 binding motifs, we point mutated C69 (C69A) and C87 (C87U) in VTRNA1.1 and performed RNA pull-down experiments (Fig. S1B)."

>> Why was position C87 chosen? Also, in the accompanying Fig. S1B it is C88 not C87 that was mutated?

> Page 7: "...the amount of VTRNA1.1 bound to SRSF2 was highest in the absence of NSUN2 (Fig. 2D)."

>> The RNA-IP experiment was normalised to a RIP with an unrelated antibody (Antibody ctrl). Quantification shows that vtRNA1.1 appears to be enriched by a factor of 1.5 to 2.5 when pulling on SRSF2 over the ctrl. in NSun2 +/+ or NSun2 +/- cells. From this it seems that a lot of VTRNA1.1 is sticking to the antibody beads begging the question how much difference SRSF2 binding really makes. The increase to 4-fold enrichment over ctrl in the NSun2 -/- is correctly stated but since all presented data point to a factor of 2, which defines the difference of SFSR2 binding to un-methylated/methylated it would be really important to add data about the methylation status of VTRNA1.1 in the used cell line.

> Page 7: "To determine where VTRNA1.1 processing took place, we purified nuclear and cytoplasmic extracts from NSUN2-/- cells and measured the abundances of svRNA1 and 4."

>> On page 5 it was stated: "...because only the wild-type construct of NSUN2 increased svRNA4 production (Fig. 1B)". It is unclear why the authors decided to just fractionate NSun2-/- cells rather than also NSun2 +/+ cells and it would be helpful to do so and to test the levels of VTRNA1.1 along with the processed small RNAs in these subcellular fractions.

> Page 8: "RNA pull-down assays using biotinylated VTRNA1.1 confirmed binding of both SRSF2 and DROSHA but not DICER with higher affinity to un-methylated VTRNA1.1 using a non-stringent low ionic buffer (Fig. 2G)."

>> The presented western blot does not allow to make such strong statement because of the

unequal signal of the used RIP control (hnRNPA1), which shows clearly a stronger signal for un-methylated RNA bait. It is advisable to use either a more stringent buffer system or changes in other parameters in order to determine the binding of Drosha to the SRSF2-RNA complex.

> Page 8: "The processing of VTRNA1.1 into svRNA4 was significantly increased when SRSF2 was repressed, while the processing into svRNA1 was unaffected by down-regulation of SRSF1 and 2 (Fig. 2J)."

>> This is one of the moments in the flow of the paper where this referee has conceptual problems with the system. It would help other readers to clarify how an identical RNA species (VTRNA1.1) can give rise to two overlapping small RNAs (svRNA1 and 4) but binding to the parent molecule by SRSF2 inhibits the processing into one (svRNA4) but not the other.

> Page 9: "Thus, epidermal progenitors are an excellent model to study the cellular functions of svRNA4, due to high expression of NSUN2 and consequently high levels of RNA methylation."

>> The authors should provide methylation data from epidermal progenitors to substantiate the claim of "consequently high levels of RNA methylation".

> Page 10: "Finally, we confirmed that the svRNA4-transduced keratinocytes failed to undergo morphological changes that are normally accompanied with differentiation into a stratified layers (Fig. 3I)".

>> Is a DIC image sufficient for this claim? Can such stratification into layers be shown with Ab stainings?

> Page 10: "...and that the svRNA4 mimic reduced the colony forming efficiency of keratinocytes, which measures the capacity of single cells to form colonies (Fig. S3A)."

>> The colony formation assay needs to be quantified correctly by stating how many colonies were counted per treatment and cell type.

> Page 10: "svRNAs are regulatory small non-coding RNAs, and we previously computationally predicted potential svRNA4-target mRNAs (6,23,45)."

>> Only reference (6) contains such prediction data.

> Page 10: "Indeed, Ovo1 was efficiently repressed in the presence of svRNA4 on both RNA and protein levels (Fig. 3E, H)."

>> Although not crucially important for the manuscript one wonders how svRNA4 is acting on Ovo1 RNA. Hussain et al., 2013 have shown association of svRNA4 with Argonautes. Since the authors could not provide the identity of the nuclease processing VTRNA1.1 into svRNA4 it would improve the story to add such detail, for instance by siRNA-mediated k.d. of the predicted Argonaute mRNAs and measuring the stability of Ovo1 mRNA.

> Page 11: "As expected, the human keratinocytes underwent efficient differentiation and only Ovo1 RNA levels significantly increased when compared to cells transfected with the control siRNA construct (Fig. 4F)."

>> That Ovo1 RNA levels significantly increased is a statement that is not supported by statistical evaluation. From the provided standard deviation this increase is hard to discern and more biological replicates are needed to clarify that statement.

> Page 11: "However, down-regulation of SRSF2 dramatically affected cell division and arrested the keratinocytes in the G2/M phase of the cell cycle (Fig. 4G, H)."

>> This is an overstatement because a G2/M phase arrest looks different. The data show merely an increase of G2/M phase cells but the G0/G1 peak is equally populated in ctrl and SRSF2 siRNA treatment.

> Page 11: "...leading to reduced colony forming efficiency (Fig. 1, left and middle panels)."

>> The picture is clear but such a colony formation assay needs to be quantified correctly by

stating how many colonies were counted per treatment and cell type, especially when later stating: "...because simultaneous deletion of NSUN2 and SRSF2 only slightly rescued the colony forming efficiency of the human keratinocytes (Fig. 4I, J)." If the reader counts using the provided images, then the middle panel in Fig. 4I shows 4 large colonies and the right panel shows 6 large colonies. This referee is not convinced that this can be the basis for a scientific statement.

> Page 12: "Our study provides one of the first large scale quantitative proteomics approaches to identify novel RNA binding proteins involved in 'reading' the m5C modification."

>> This is a misleading overstatement and needs to be corrected because the authors have only identified two proteins that barely bind differentially to methylated versus un-methylated RNA when performing two replicate experiments (one data set: not shown).

Furthermore, the authors did not even analyse the protein supposedly binding to m5C in RNA (PUS7) but chose the protein that does rather bind to un-methylated RNA (SRSF2).

This referee suggests to work on the discussion to take away some of the potential confusion a reader might be experiencing when trying to conceptualise what the methyl group at C69 in VTRNA1.1 might be attracting or repelling depending on the differentiation state of the cell.

Reviewer #2 (Remarks to the Author):

In this manuscript Sajini et al. studied the molecular consequences of cytosine methylation of vault RNA 1.1 (vtRNA1.1) both in vitro and in vivo. Using SILAC labeling and mass spectrometry the authors identify a panel of proteins with preferential binding to methylated or un-methylated vtRNA1.1, respectively. The splicing factor SRSF2 that was enriched for preferential binding to the un-methylated vtRNA1.1 was chosen for further analysis, and binding of SRSF2 was confirmed by Western blotting of pull-down experiments using methylated or un-methylated vtRNA1.1 as a bait. To determine interaction sites for SRSF2 on vtRNA1.1 the authors mutated one nucleotide each in two putative binding motifs and found a decrease in SRSF2 affinity by RNA pull-down. Conversely, interaction between SRSF2 and vtRNA1.1 was highest in cells lacking Nsun2. The authors then went on to determine if vtRNA1.1 processing occurred in the nucleus or cytoplasm and which nuclease might be involved. It is not clear though, that the results from these experiments support the conclusions (see comments below). To demonstrate a role for SRSF2 in vtRNA1.1 processing, SRSF2 was knocked down resulting in an upregulation of svRNA4 but unchanged svRNA1 levels indicating that SRSF2 prevents production of svRNA4.

To study the cellular effects of vtRNA1.1 processing, differentiation of primary human keratinocytes was used as a model system. Terminally differentiated cells showed decreased Nsun2 and svRNA4 but increased vtRNA1.1 levels. Overexpression of svRNA4 reduced expression of differentiation markers, and inhibition of endogenous svRNA4 (and vtRNA1.1) induced differentiation in this system indicating a role for svRNA4 in maintaining a committed but undifferentiated state of the progenitor cells. Further experiments attempt to link the observed regulation by svRNA4 to a direct targeting of the transcription factor OVOL1. SRSF2 knock-down and cell differentiation experiments resulted in a change in cell cycle profile of differentiating cells and reduced colony formation efficiency. Simultaneous knock-down of SRSF2 and Nsun2 rescued the low svRNA4 levels of Nsun2-depleted cells to wild-type levels during differentiation.

In summary, this manuscript presents very interesting new findings with respect to the interpretation of RNA cytosine methylation by protein factors, and it sheds new light on the cellular function of the elusive vault RNAs. Some issues need to be addressed with additional experiments to assure that the claims regarding SRSF2/vtRNA1.1 interaction, the processing of vtRNA1.1, and the cellular role of svRNA4 are justified.

Major issues

1. Figure 1B: The data show that in cells lacking the RNA methyltransferase Nsun2, processing of vtRNA1.1 into small vault RNA svRNA4 is reduced and cannot be rescued by expression of catalytically inactive Nsun2. Even expression of wild-type Nsun2 is not able to fully rescue svRNA4 levels to those of Nsun2^{+/-} cells. The authors should show protein levels of Nsun2 in the transfected cell lines to determine if svRNA4 processing is related to expression levels of Nsun2 (mutant) protein. Also, details about transfection constructs used in this assay are missing in the methods section. Furthermore, clarification is needed, whether Nsun2^{+/-} or Nsun2^{+/+} cells were used. The Figure says Nsun2^{+/-}, while the text on page 5, line 105, states Nsun2^{+/+}. Statistical significance should also be determined for Nsun2^{-/-} versus Nsun2^{-/-} (Nsun2wt).

2. Fig. 1D. Please explain how the numbers were calculated in the upper panel (as well as in Suppl. Fig. 1D). The data indicate a two-fold enrichment of SRSF2 on non-methylated vtRNA1.1. However, in the control pull-down with hnRNPA1, the band with un-methylated vtRNA1.1 appears stronger, too, and this is similar in Supplementary Figure S1D. If hnRNPA1 is used to normalize for unspecific binding, the signals should not be saturated for quantification.

3. Fig. 1E and Supplementary Fig. S1CD: The pull-down experiments indicate that the two putative SRSF2 binding motifs on vtRNA1.1 are indeed involved in SRSF2 binding. Yet, some important issues have not been addressed. For example, use of RNA pull-down with cell lysates does not prove direct binding of SRSF2 to vtRNA1.1. It is not shown, whether SRSF2 needs both motifs to bind or if the two motifs are bound by separate molecules of SRSF2. Finally, binding affinities for methylated versus un-methylated vtRNA1.1 could be determined more accurately. Use of a complementary experimental approach, such as EMSA, may be suitable to answer some of these questions. Since one of the main findings of this work is the binding of SRSF2 to un-methylated vtRNA1.1, closer investigation of this issue is needed.

4. Fig. 2E, F, G: The results in Fig. 2E suggest that svRNA1 and svRNA4 show different relative abundance in nuclear and cytoplasmic fractions. They do not, however, allow for the conclusion that processing occurs in the nucleus, since svRNA4 might be transported back into the nucleus after processing in the cytoplasm.

Moreover, I would guess that the normalizer for qPCR mir-16 is not present at similar levels in nuclear and cytoplasmic extracts (2E). Therefore, unless the absolute amounts of mir-16 is determined in both fractions and considered in the normalization, it is not possible to do statistics on the comparison between nuclear and cytoplasmic extracts.

From Fig. 2F it appears that svRNA4 processing is stronger than svRNA1 processing using a nuclear extract. The same experiment needs to be performed with a cytoplasmic extract.

Depending on the results, it may be possible to conclude as to the location of vtRNA1.1 processing. Likewise, to demonstrate a dependence of svRNA4 processing on C69 methylation, the experiments should also be carried out with methylated vtRNA1.1.

The binding data for DROSHA are very weak. If the authors wish to claim that DROSHA is the responsible nuclease, they need to knock-down DROSHA and monitor the consequences for vtRNA1.1 processing. Otherwise, Fig. 2G should be deleted.

5. Fig. 2J: How do the authors explain increased svRNA4 but unchanged svRNA1 levels upon SRSF2 RNAi? One would expect that at constant vtRNA1.1 levels, increased svRNA4 should cause decreased svRNA1. What are the vtRNA1.1 levels under these conditions?

6. Fig. 3 and Supplementary Fig. S3. The authors conclude from the experiments shown here that svRNA4 targets *Ovo1* mRNA for repression. While this scenario is entirely possible, the provided evidence (computational prediction of svRNA4 binding site in *Ovo1* mRNA and *Ovo1* knock-down experiment), however, does not strictly prove such a direct connection. Unless the authors are

willing to perform a mutation analysis of the putative svRNA4 binding sites in Ovol1 mRNA, the respective statements on page 10, lines 238-239, and in the abstract should be toned down.

The CFE results in Fig. S3A are not suitable to draw any conclusions. Please, show quantification. Also, there seems to be an error in the legend, as it states that the cells were transfected with svRNA mimic when presumably it should read Ovol1 siRNA.

7. Fig. 4: Figs. 4D and F are confusing. Why are Srsf1 levels not reduced upon Srsf1-RNAi (4D)? Why is expression for marker genes vtRNA and Krt10 shown for Srsf1 but not Srsf2 knock-down (4F)? What is the purpose of the Srsf1 knock-down in this experiment? What is the p value for Ovol1?

Fig. 4G, H: The results do not suggest a "dramatic" effect on cell division as stated on page 11, line 255.

Fig. 4J: A quantification of CFE of control cells should be included for comparison.

Fig. 4K: Why does svRNA4 seem to be reduced in Srsf2 knock-down cells? The opposite would be expected.

Minor

Page 11, line 256: Fig. 4I instead of Fig. I

Page 20: Can the authors explain, why they use ATP and creatine-phosphate in the RNA-mediated protein pull-down assays?

Page 29: In the legend to Fig. 1 the term "agarose labelled methylated or un-methylated C69" is used. Do the authors mean "agarose-coupled"?

What is the control in Fig. 3E?

In Fig. 3C, why was svRNA1 expression not tested?

In Fig. 5A, B, C: cytoplasm is misspelled.

Page 6, line 139, page 11, lines 257,258: change SFSR2 to SRSF2

NCOMMS-18-02777

We consider our revised manuscript to have been tremendously strengthened by addressing the reviewers' feedback, and we thank the referees for the very constructive criticism.

In response to the referees's comments, we have made the following changes to the manuscript:

Reviewer #1

NSun2 is a (cytosine-5) methyltransferase that acts mostly on tRNAs. A few alternative substrates, most of them non-coding RNAs, have been identified using transcriptome-wide approaches such as miCLiP (the authors of the presented manuscript), Aza-IP and RNA bisulfite sequencing. This referee is aware of a manuscript by Xin Yang et al., Pubmed: 28418038, which recently reported on robust NSun2-dependent messenger RNA methylation in HeLa cells. However, since it was also reported that many cancer cell lines (including HeLa cells) show genomic gains of the NSun2 locus, it stands to reason that a number of the identified m5C sites in these cancer cell lines are not physiological but are in fact artefacts caused by RNA methyltransferase overexpression.

The referee is correct. It is well reported that cancer cell lines often over-express the NSUN2 protein due to a genomic amplification of chromosome 5p15, on which the NSUN2 gene is located (Frye et al., 2009).

However, this study is not performed in HeLa or HEK293 cells. We only used HeLa cells for the quantitative mass spectrometry analyses. All other experiments were performed in either normal human dermal fibroblasts or patient-derived dermal fibroblast (hDF) lacking the functional NSUN2 gene (Martinez et al., 2012), or in healthy primary human keratinocytes (HK).

To demonstrate the physiological relevance of the methylated site, we now include RNA bisulfite (BS) sequencing data on human embryonic stem cells (H9) and primary human keratinocytes to demonstrate that C69 in VTRNA1.1 is consistently methylated (see Fig. 1D, E and Fig. 4G,H).

Importantly, we do not find any evidence that over-expression of NSUN2 leads to unspecific RNA methylation.

This notion has to be taken also into account when performing experiments in HEK293 cells, the cell system used for the identification of vault RNA (vtRNA) methylation by NSun2 as reported by Hussain et al., 2013. In this paper, two out of four vtRNAs contained m5C as confirmed by targeted RNA bisulfite sequencing. Interestingly, loss of NSun2-mediated RNA methylation on only one of these vtRNAs (vtRNAs 1.1) affected its processing into smaller RNAs (svRNAs). And rather counter-intuitively, and contrary to the one-sided effects of NSun2-mediated RNA methylation on tRNAs, loss of NSun2 both increased and decreased the stability of vtRNAs1.1 as measured by the abundance of 4 different small RNAs that were all derived from vtRNAs1.1.

The referee is correct, in *Hussain et al. 2013* we used HEK293 cells to perform the miCLIP assay. Yet, the RNA bisulfite sequencing was performed in the human dermal fibroblasts obtained from the patients (see also comment above). VTRNA1.1 was one of the non-coding RNAs confirmed to be methylated by NSUN2 in both assays and in both cell lines.

The author is further correct that the abundance of svRNA4, contrary to tRNA-derived non-coding RNAs, correlated with the presence of NSUN2 and accordingly methylation of C69. That was one of the reasons, why we chose to analyse this particular svRNA in greater detail. However, similar to the function of m⁵C in tRNAs, we find that the function of m⁵C in VTRNA1.1 is also to repel RNA-binding proteins.

General comments:

> *The wording of the manuscript title indicates a seemingly wide and review-like scope of the work. The title should be changed to a more pointed statement because the manuscript actually does not address the writing and reading of 5mC in RNA but rather that a lack of 5mC on vtRNAs is determining its processing.*

We agree with the referee and changed the title to ‘*Loss of 5-methylcytosine alters the biogenesis of Vault-derived small RNAs to coordinate epidermal differentiation*’.

> *The reasoning for the experimentation, especially after the initial identification of proteins binding to un-methylated versus methylated vtRNAs1.1 is rather convoluted and often hard to follow. This referee suggests restructuring into more than the existing 4 subheadings.*

According to the referee’s recommendation we have restructured the text into more subheadings (see Result section).

> *Given the introduction to m⁵C, NSun2 and the potential biological function of the established RNA methylation circuits, it remains largely unclear to the reader why the authors chose to study a protein (SRSF2) that does un-methylated RNA rather than methylated RNA.*

Both the presence and absence of site-specific m⁵C modulates the affinity to RNA-binding proteins (Blanco et al., 2014). Therefore, studying proteins binding specifically the methylated or un-methylated RNA is equally important, because their balance and interplay will determine the consequences on the RNA activity.

The importance of sites lacking m⁵C modifications is highlighted by our finding that only the absence of m⁵C in transfer RNA (tRNAs) is functionally relevant: the absence of the site-specific m⁵C modification increases the affinity to an endonuclease, which cleaves tRNAs into small non-coding RNAs. These tRNA-derived fragments then act as highly important functional regulators in the cellular stress response (Blanco et al., 2016; Blanco et al., 2014).

We have now revised the text and also changed the abstract to clarify why we chose SRSF2 as a candidate for further analyses.

However, only after looking at the data in Figure 1C and the Figure Legend the reader realises that the only proteins that were repeatedly identified in two independent RNA pull-down experiments (asterisk) on SILAC-treated cells followed by mass spectrometry were SRSF2 and PUS7. That all other proteins, which showed a greater differential in peptide counts than SRSF2, might have bound disadvantageously to vtRNA1.1 is a major concern for the interpretation of the robustness of the used initial approach, especially in light of the fact that SRSF2 seems to bind to both un-methylated and methylated vtRNAs1.1 (Figure 1D) and that the log₂ value of the fold-change (FC) of identified peptides is 0.5 for SRSF2 and 0.7 for PUS7 in the differential RNA pull-downs. This are indeed very small values and raise the question as to the reproducibility of these initial results, which, importantly define all other experiments in the manuscript.

We apologize that our manuscript lacked clarity on the description of the SILAC pull-down assay. To demonstrate the robustness of the SILAC mass spec screen, we have now included the following information: (i) The raw data from both screens (a total of three replicates) (see Table S2 and S3), (ii) high correlation between technical replicates (Fig. S2B), and (iii) the 144 commonly identified proteins in all three replicates that bind to VTRNA1 (Fig. 2A; Fig. S2C).

Please note that we only find a small number of proteins that show a quantitatively different binding to methylated and un-methylated VTRNA1.1 (Fig. 2A). Out of these, PUS7 and SRSF2 were the most differently bound proteins.

> *VTRNA1.1* methylation was identified using HEK293 cells. However, the presented manuscript uses also HeLa cells, primary human keratinocytes and their differentiated progeny. Nowhere in the manuscript is a mention of the methylation status of *VTRNA1.1* in these cell types, only the assumption that this RNA is methylated as observed in HEK293. The authors should provide (quantitative) RNA methylation data (preferably by RNA bisulfite sequencing) where needed (see below), especially in light of a “magic” factor of 2 that is defining the binding of SRSF1 to un-methylated/methylated *VTRNA1.1*. From the original publication by Hussain et al., 2013 this referee gathers from the heat map that the methylation levels might be at around between 25 to 50% at position C69. Could that partial methylation help the authors explain why SRSF1 is binding only a little bit better to the un-methylated RNA?

VTRNA1.1 methylation at position C69 was identified in HEK293 and dermal fibroblasts derived from patients lacking a functional NSUN2 protein using miCLIP and RNA BS-sequencing respectively (Hussain et al., 2013). Except for the SILAC pull-down assay, we do not use HeLa cells for any further experiments. We have now better clarified this in the Methods section.

We now provide further evidence demonstrating the wide-spread occurrence of m⁵C in VTRNA1.1 in human cells, including embryonic stem cells and primary keratinocytes (see Fig. 1D, E; Fig 4G). Notably, we did not find a corresponding site in mouse cells. This information is now included in the text.

Moreover, we show that methylation of C69 in VTRNA1.1 is solely mediated by NSUN2 through rescue experiments using the wild-type and an enzymatic dead construct of NSUN2 (K190M) (see Fig. 1A-D).

Please note that the methylation level of C69 is never 100% and in most cases less than 50% including cells that over-express NSUN2. Thus, partial methylation might explain why SRSF2-binding is only a little bit better to the un-methylated RNA. However, it could also be explained by higher turnover (processing) of methylated VTRNA1.1.

> Please, double-check all Figure calls. For instance, on page 5: Fig. 1B is wrongly called as Fig. 2B.

We have double-checked all Figures and corrected the mistake.

> Please, double-check all Reference calls. For instance, on page 12: citation (47) is not correct in the context of the statement of the sentence.

We have double-checked all references and corrected the mistake.

> RNA scientist like to see the RNAs they study usually on Northern blots. VTRNA1.1 can be detected nicely by northern blotting (Amort et al., 2015) and this referee wonders whether or not northern blotting would substantiate some of the claims the authors make (see below).

We agree with the referee that VTRNA1.1 can be readily detected by Northern Blot, yet the svRNAs are far less abundant. To confirm the differential abundance of svRNAs, we therefore used two different and independent methods: (i) small RNA sequencing and (ii) RT-QPCR of size selected small RNA isolated from TBE gels. In addition, we believe that small RNA seq and small QPCR are the most suitable methods to distinguish svRNA4 from svRNA1 (see Fig. S1G).

> Referees should have access to primary data, especially when major conclusions are based on such data. In the case of the presented manuscript, there is no indication what the read counts from the mass spec data are, which is important to judge certain Figures submitted with the manuscript.

We agree with the referee and sincerely apologize for the oversight. The raw data from the mass spec are now provided as Supplementary Tables (see Tab. S2 and S3).

Specific comments:

> Page 5: “The processing of VTRNA1.1. into svRNA4 depended on the methylation activity of NSUN2 because only the wild-type construct of NSUN2 increased svRNA4 production (Fig. 1B).” -- To claim that methylation activity is involved the authors should provide RNA bisulfite data on VTRNA1.1 for each experiment to show that NSun2^{-/-} cells regain m5C when overexpressing wt NSun2.

As the referee requested, we now provide RNA bisulfite data for each experiment (see Fig. 1A-E and Fig. S1A-E).

> Page 5: "Thus, the presence of a methylation group at C69 determined the processing of VTRNA1.1 into svRNA4." -- It would be helpful to show Northern data for the 29 nt small svRNA4 to substantiate the claim that this small RNA is being produced in a methylation-dependent fashion.

To confirm the differential abundance of svRNAs, we used two different and independent methods: (i) small RNA sequencing and (ii) RT-QPCR of size selected small RNA using TBE gels. The primers are chosen in such a way that we excluded all other svRNAs.

> Page 6: "...we performed RNA pull-down assays, and confirmed that SRSF2 bound non-methylated (C69) with higher affinity than methylated (m5C69) VTRNA1.1 (Fig. 1D)."

>> Since the authors prepared the RNA bait themselves using activated agarose and have not provided data about the efficiency of RNA coupling it remains unclear whether similar molarities of methylated and un-methylated vtRNA1.1 have been subjected to the assay. Therefore, it is wrong to call the presented data measured affinities between RNA and protein because SRSF2 protein was probed by western blotting. If the authors would like to keep the wording "affinities" they need to present data containing experiments with equimolar SRSF2 protein concentrations on equimolar mass of bait RNA. If the authors cannot contribute such data, they need to tone down the scientific description of the data.

To substantiate our claims, we now additionally provide EMSA experiments that confirm our finding that SRSF2 binds un-methylated VTRNA1.1 with higher affinity (Fig. 2E, F)

> Page 6: "hnRNP A1 protein served as a control as it displayed equal binding to m5C69 and C69 VTRNA1.1." -- The authors should mention that hnRNPA1 binds to vtRNA1.1. Either by referring to previous published data or to their own data from the SILAC pull-down experiments.

We have now included the reference.

> Page 6: "VTRNA1.1 contains two putative SFSR2 RNA binding motifs (RRM1 and RRM2), one of which overlapped with the methylated cytosine 69 (Fig. 1E)." -- The authors should provide a reference as to how such SFSR2 binding motifs are defined.

We have now included the references.

> Page 6: "To validate the functional importance of the SRSF2 binding motifs, we point mutated C69 (C69A) and C87 (C87U) in VTRNA1.1 and performed RNA pull-down experiments (Fig. S1B)." -- Why was position C87 chosen? Also, in the accompanying Fig. S1B it is C88 not C87 that was mutated?

We have now corrected this mistake.

> Page 7: “...the amount of VTRNA1.1 bound to SRSF2 was highest in the absence of NSUN2 (Fig. 2D).” -- The RNA-IP experiment was normalised to a RIP with an unrelated antibody (Antibody ctrl). Quantification shows that vtRNA1.1 appears to be enriched by a factor of 1.5 to 2.5 when pulling on SRSF2 over the ctrl. in NSun2 +/+ or NSun2 +/- cells. From this it seems that a lot of VTRNA1.1 is sticking to the antibody beads begging the question how much difference SRSF2 binding really makes. The increase to 4-fold enrichment over ctrl in the NSun2 -/- is correctly stated but since all presented data point to a factor of 2, which defines the difference of SRSF2 binding to un-methylated/methylated it would be really important to add data about the methylation status of VTRNA1.1 in the used cell line.

As requested by the referee, we now show the methylation status in the used cell lines (see Fig. 1C-E; Fig. 4G,H; Fig. S1B-E; Fig. S4A, B).

> Page 7: “To determine where VTRNA1.1 processing took place, we purified nuclear and cytoplasmic extracts from NSUN2-/- cells and measured the abundances of svRNA1 and 4.” -- On page 5 it was stated: “...because only the wild-type construct of NSUN2 increased svRNA4 production (Fig. 1B)”. It is unclear why the authors decided to just fractionate NSun2-/- cells rather than also NSun2 +/+ cells and it would be helpful to do so and to test the levels of VTRNA1.1 along with the processed small RNAs in these subcellular fractions.

We decided to only fractionate NSUN2-/- cell lysates because these cells have the lowest abundance of svRNA4 and we reasoned that this would decrease background levels coming from endogenous svRNA4 for the cleavage assay. We have now clarified this in the text.

> Page 8: “RNA pull-down assays using biotinylated VTRNA1.1 confirmed binding of both SRSF2 and DROSHA but not DICER with higher affinity to un-methylated VTRNA1.1 using a non-stringent low ionic buffer (Fig. 2G).” -- The presented western blot does not allow to make such strong statement because of the unequal signal of the used RIP control (hnRNPA1), which shows clearly a stronger signal for un-methylated RNA bait. It is advisable to use either a more stringent buffer system or changes in other parameters in order to determine the binding of Drosha to the SRSF2-RNA complex.

We agree with the referee and since our initial conclusion was that neither SRSF2 nor NSUN2 were part of the Drosha-RNA complex, we omitted these experiments from our study.

> Page 8: “The processing of VTRNA1.1 into svRNA4 was significantly increased when SRSF2 was repressed, while the processing into svRNA1 was unaffected by down-regulation of SRSF1 and 2 (Fig. 2J).” -- This is one of the moments in the flow of the paper where this referee has conceptual problems with the system. It would help other readers to clarify how an identical RNA species (VTRNA1.1) can give rise to two overlapping small RNAs (svRNA1 and 4) but binding to the parent molecule by SRSF2 inhibits the processing into one (svRNA4) but not the other.

Please note that the production of svRNA4 precludes the formation of svRNA1 because the fragments are overlapping (Fig. S1G). Indeed, we observed a down-regulation of svRNA1 upon removal of SRSF2, yet this change was not significant (Fig. 3G). We also consistently measured higher levels svRNA1 than svRNA4, which could be due to a greater stability of svRNA1; and therefore, changes in its abundances might be more difficult to quantify.

> Page 9: “Thus, epidermal progenitors are an excellent model to study the cellular functions of svRNA4, due to high expression of NSUN2 and consequently high levels of RNA methylation.” >>The authors should provide methylation data from epidermal progenitors to substantiate the claim of “consequently high levels of RNA methylation”.

As the referee requested, we now provide RNA bisulfite data for undifferentiated and differentiated primary HK (Fig. 4D-H and Fig. S4A, B).

> Page 10: “Finally, we confirmed that the svRNA4-transduced keratinocytes failed to undergo morphological changes that are normally accompanied with differentiation into a stratified layers (Fig. 3I).”>> Is a DIC image sufficient for this claim? Can such stratification into layers be shown with Ab stainings?

We agree with the referee and therefore provide quantifications of well-established terminal differentiation markers on RNA and protein level using Western blotting and RT-QPCR (see Fig.4J-L) (Watt, 1989).

> Page 10: “...and that the svRNA4 mimic reduced the colony forming efficiency of keratinocytes, which measures the capacity of single cells to form colonies (Fig. S3A).” >> The colony formation assay needs to be quantified correctly by stating how many colonies were counted per treatment and cell type.

We have now removed former Supplementary Figure 3A from the manuscript.

> Page 10: “svRNAs are regulatory small non-coding RNAs, and we previously computationally predicted potential svRNA4-target mRNAs (6,23,45).” >> Only reference (6) contains such prediction data.

This mistake has been corrected.

> Page 10: “Indeed, *Ovol1* was efficiently repressed in the presence of svRNA4 on both RNA and protein levels (Fig. 3E, H).” >> Although not crucially important for the manuscript one wonders how svRNA4 is acting on *Ovol1* RNA. Hussain et al., 2013 have shown association of svRNA4 with Argonautes. Since the authors could not provide the identity of the nuclease processing *VTRNA1.1* into svRNA4 it would improve the story to add such detail, for instance by siRNA-mediated k.d. of the predicted Argonaute mRNAs and measuring the stability of *Ovol1* mRNA.

We agree with the referee that the experiment was not crucially important for our study and we have omitted the experiments from the revised version of the manuscript.

> Page 11: “As expected, the human keratinocytes underwent efficient differentiation and only *Ovol1* RNA levels significantly increased when compared to cells transfected with the control siRNA construct (Fig. 4F).” >> That *Ovol1* RNA levels significantly increased is a statement that is not supported by statistical evaluation. From the provided standard deviation this increase is hard to discern and more biological replicates are needed to clarify that statement.

We have now included the significance measures into all figures whenever we measured statistically significant differences.

> Page 11: “However, down-regulation of *SRSF2* dramatically affected cell division and arrested the keratinocytes in the G2/M phase of the cell cycle (Fig. 4G, H).” >> This is an overstatement because a G2/M phase arrest looks different. The data show merely an increase of G2/M phase cells but the G0/G1 peak is equally populated in ctrl and *SRSF2* siRNA treatment.

To provide more convincing evidence that *SRSF2* is required for cell division in primary human keratinocytes, we now included an additional experiment, in which we transfected the undifferentiated epidermal cells with the *SRSF2* siRNA (Fig. S5C). Removal of *SRSF2* not only led to the down-regulation of major cell cycle regulators, it also caused cell death (see Fig. 5D-H).

> Page 11: “...leading to reduced colony forming efficiency (Fig. I, left and middle panels).” >> The picture is clear but such a colony formation assay needs to be quantified correctly by stating how many colonies were counted per treatment and cell type, especially when later stating: “...because simultaneous deletion of *NSUN2* and *SRSF2* only slightly rescued the colony forming efficiency of the human keratinocytes (Fig. 4I, J).” If the reader counts using the provided images, then the middle panel in Fig. 4I shows 4 large colonies and the right panel shows 6 large colonies. This referee is not convinced that this can be the basis for a scientific statement.

The main conclusions from this experiment were that *SRSF2* is essential for keratinocyte cell division and survival, and that this function was largely independent of the methylation-dependent processing of *VTRNA1.1*. Therefore, we have replaced the experiments with data showing that *SRSF2* is required for cell division and survival (see Fig. 5D-G).

> Page 12: “Our study provides one of the first large scale quantitative proteomics approaches to identify novel RNA binding proteins involved in ‘reading’ the m5C modification.” >> This is a misleading overstatement and needs to be corrected because the authors have only identified two proteins that barely bind differentially to methylated versus un-methylated RNA when performing two replicate experiments (one data set: not shown).

Furthermore, the authors did not even analyse the protein supposedly binding to m⁵C in RNA (PUS7) but chose the protein that does rather bind to un-methylated RNA (SRSF2). This referee suggests to work on the discussion to take away some of the potential confusion a reader might experiencing when trying to conceptualise what the methyl group at C69 in VTRNA1.1 might be attracting or repelling depending on the differentiation state of the cell.

We respectfully disagree with the referee that we only identified two proteins that barely bind differentially to methylated versus un-methylated RNA. We now provide the full data set showing that we identified 144 proteins that bind to VTRNA1.1 in two independent experiments (see Fig. 2A and Fig. S2C). It is unclear to us why the referee expects to find a large number of proteins that bind differentially to unmethylated and methylated VTRNA1.1 that carries the m⁵C mark at one distinct cytosine at position C69.

Protein reading m⁵C and being repelled by the methyl group are equally important to determine subsequent function and activity of a targeted RNA. As the referee requested, we have re-written the discussion to clarify this point.

Reviewer #2:

In summary, this manuscript presents very interesting new findings with respect to the interpretation of RNA cytosine methylation by protein factors, and it sheds new light on the cellular function of the elusive vault RNAs. Some issues need to be addressed with additional experiments to assure that the claims regarding SRSF2/vtRNA1.1 interaction, the processing of vtRNA1.1, and the cellular role of svRNA4 are justified.

We thank the referee for pointing out that our study provides very interesting new findings with respect to the interpretation of RNA cytosine methylation by protein factors.

To address this referee's concerns, we have now provided the following additional experiments: (i) We now provide an EMSA assay confirming that SRSF2 preferentially binds unmethylated VTRNA1.1, (ii) we determined the methylation status of VTRNA1.1 in all analysed cell systems, and (iii) provide further evidence for the essential cell functions of SRSF2 in primary human keratinocytes (please see our detailed responses below).

Major issues

1. Figure 1B: The data show that in cells lacking the RNA methyltransferase Nsun2, processing of vtRNA1.1 into small vault RNA svRNA4 is reduced and cannot be rescued by expression of catalytically inactive Nsun2. Even expression of wild-type Nsun2 is not able to fully rescue svRNA4 levels to those of Nsun2^{+/-} cells. The authors should show protein levels of Nsun2 in the transfected cell lines to determine if svRNA4 processing is related to expression levels of Nsun2 (mutant) protein. Also, details about transfection constructs used in this assay are missing in the methods section. Furthermore, clarification is needed, whether Nsun2^{+/-} or Nsun2^{+/+} cells were used. The Figure says Nsun2^{+/-}, while the text on

page 5, line 105, states *Nsun2*^{+/+}. Statistical significance should also be determined for *Nsun2*^{-/-} versus *Nsun2*^{-/-} (*Nsun2*^{wt}).

We thank the referee for raising this important issue. We now included data showing NSUN2 levels in all the conditions analyzed and confirm equal expression of the over-expressed constructs (Fig. 1H; Fig. S1A). With regard to the C271A and C321A lines, these are precisely the same stably infected cell lines from our previous publication (Blanco et al., 2016), where we demonstrated equal protein expression of the constructs by Western Blot (*Extended Data Figure 9c*). We have now clarified this in the text.

We have now included the details about the transfection constructs used in this assay into the Methods section.

We have corrected the text according to which cell line was used.

We have now included the missing statistical significance into former Figure 1D.

2. Fig. 1D. Please explain how the numbers were calculated in the upper panel (as well as in Suppl. Fig. 1D). The data indicate a two-fold enrichment of SRSF2 on non-methylated *vtRNA1.1*. However, in the control pull-down with *hnRNPA1*, the band with un-methylated *vtRNA1.1* appears stronger, too, and this is similar in Supplementary Figure S1D. If *hnRNPA1* is used to normalize for unspecific binding, the signals should not be saturated for quantification.

We used ImageJ to quantify Western blots following the manual's instructions. The intensity of each band was normalized to the intensity of its individual loading control. Thus, stronger bands in the loading control are accounted for.

3. Fig. 1E and Supplementary Fig. S1CD: The pull-down experiments indicate that the two putative SRSF2 binding motifs on *vtRNA1.1* are indeed involved in SRSF2 binding. Yet, some important issues have not been addressed. For example, use of RNA pull-down with cell lysates does not prove direct binding of SRSF2 to *vtRNA1.1*. It is not shown, whether SRSF2 needs both motifs to bind or if the two motifs are bound by separate molecules of SRSF2. Finally, binding affinities for methylated versus un-methylated *vtRNA1.1* could be determined more accurately. Use of a complementary experimental approach, such as EMSA, may be suitable to answer some of these questions. Since one of the main findings of this work is the binding of SRSF2 to un-methylated *vtRNA1.1*, closer investigation of this issue is needed.

We now confirmed direct binding of SRSF2 to *VTRNA1.1* by including an EMSA using GST-labelled SRSF2 (see Fig. 2E, F).

We had indeed generated a construct carrying mutations at both binding motifs. However, the results were inconclusive. We reasoned that both mutations together may influence *VTRNA1.1* folding and therefore, omitted the data from our study.

4. Fig. 2E, F, G: The results in Fig. 2E suggest that *svRNA1* and *svRNA4* show different relative abundance in nuclear and cytoplasmic fractions. They do not, however, allow for the conclusion that processing occurs in the nucleus, since *svRNA4* might be transported back into the nucleus after processing in the cytoplasm.

Moreover, I would guess that the normalizer for qPCR mir-16 is not present at similar levels in nuclear and cytoplasmic extracts (2E). Therefore, unless the absolute amounts of mir-16 is determined in both fractions and considered in the normalization, it is not possible to do statistics on the comparison between nuclear and cytoplasmic extracts.

From Fig. 2F it appears that svRNA4 processing is stronger than svRNA1 processing using a nuclear extract. The same experiment needs to be performed with a cytoplasmic extract. Depending on the results, it may be possible to conclude as to the location of vtRNA1.1 processing. Likewise, to demonstrate a dependence of svRNA4 processing on C69 methylation, the experiments should also be carried out with methylated vtRNA1.1.

We agree with the referee and now show the relative proportion of svRNA4 to svRNA1 (see Fig. 3E). We have re-written this paragraph accordingly.

We now also include the data from the cytoplasmic extract showing no significant difference in svRNA1 and 4 production (see Fig. 3D; right hand panel).

In addition, we now include data showing more svRNA4 was produced when we used nuclear extract on methylated VTRNA1.1 (see Fig. S3C). Yet, this change was not statistically significant.

The binding data for DROSHA are very weak. If the authors wish to claim that DROSHA is the responsible nuclease, they need to knock-down DROSHA and monitor the consequences for vtRNA1.1 processing. Otherwise, Fig. 2G should be deleted.

We agree with the referee and since our conclusion was that neither SRSF2 nor NSUN2 were part of the Drosha-RNA complex, we omitted the experiments from our study.

5. Fig. 2J: How do the authors explain increased svRNA4 but unchanged svRNA1 levels upon SRSF2 RNAi? One would expect that at constant vtRNA1.1 levels, increased svRNA4 should cause decreased svRNA1. What are the vtRNA1.1 levels under these conditions?

We do observe a down-regulation of svRNA1 upon removal of SRSF2, yet this change was not significant (see Fig. 3G). We consistently measured higher levels svRNA1 than svRNA4. This might be due to a greater stability of svRNA1 and therefore, changes in abundances might be more difficult to quantify.

Deletion of SRSF2 can cause cell death and human dermal fibroblasts are highly sensitive to SRSF2-deletion. Therefore, we compared the relation of full length VTRNA1.1 to svRNA4 in human keratinocytes, which better tolerate deletion of SRSF2 in the short term (see Fig. 4B, C). We find that down-regulation of svRNA4 coincides with up-regulation of VTRNA1.1.

6. Fig. 3 and Supplementary Fig. S3. The authors conclude from the experiments shown here that svRNA4 targets Ovoll mRNA for repression. While this scenario is entirely possible, the provided evidence (computational prediction of svRNA4 binding site in Ovoll mRNA and Ovoll knock-down experiment), however, does not strictly prove such a direct connection. Unless the authors are willing to perform a mutation analysis of the putative svRNA4 binding

sites in *Ovol1* mRNA, the respective statements on page 10, lines 238-239, and in the abstract should be toned down.

We agree with the referee and since the experiment was not crucially important for our study we omitted these data from the revised version of the manuscript and changed the text and abstract accordingly.

The CFE results in Fig. S3A are not suitable to draw any conclusions. Please, show quantification. Also, there seems to be an error in the legend, as it states that the cells were transfected with svRNA mimic when presumably it should read *Ovol1* siRNA.

We agree and have now removed former Supplementary Figure 3A from the manuscript.

7. Fig. 4: Figs. 4D and F are confusing. Why are *Srsf1* levels not reduced upon *Srsf1*-RNAi (4D)? Why is expression for marker genes *vtRNA* and *Krt10* shown for *Srsf1* but not *Srsf2* knock-down (4F)? What is the purpose of the *Srsf1* knock-down in this experiment? What is the *p* value for *Ovol1*? Fig. 4G, H: The results do not suggest a “dramatic” effect on cell division as stated on page 11, line 255. Fig. 4J: A quantification of CFE of control cells should be included for comparison.

Fig. 4K: Why does svRNA4 seem to be reduced in *Srsf2* knock-down cells? The opposite would be expected.

We apologize, the former Figures 4D and F were mis-labelled and should have said *Srsf2*. We have now changed the Figures and the mistake is corrected (see Fig. 5C; Fig. S5A).

The *p*-value for *Ovol1* is now included (see Fig. 5C).

To substantiate our claim that SRSF2 is required for cell division and survival, we provide additional experiments showing that knock-down of SRSF2 caused down-regulation of major cell cycle regulators (see Fig. 5E) and cell death (see Fig. 5F,G).

The CFE assay showed that SRSF2’s function on cell cycle and cell survival was upstream of NSUN2 and svRNA4 biogenesis. We have replaced the CFE assay to demonstrate that cell viability is reduced after knock-down of SRSF2, and we have re-written the Result section accordingly.

Please note that the production of svRNA4 was not significantly reduced in the absence of SRSF2 (see Fig. 5H).

Minor

Page 11, line 256: Fig. 4I instead of Fig. I

This mistake is now corrected.

Page 20: Can the authors explain, why they use ATP and creatine-phosphate in the RNA-mediated protein pull-down assays?

ATP and creatine-phosphate are the sources of energy regeneration, which is significantly depleted upon preparation of cell extracts. Moreover, many RNA-binding proteins (including RNA helicases) depend on the ATP for their normal physiological functions. Finally, upon extraction of proteins from cells the phosphatases are overpowering kinases as the cellular signalling pathways are destroyed. Many important protein phosphorylations are being lost. Thus, the addition of ATP and creatine-phosphate to cell extracts helps maintaining kinases' activity and counteracts the action of phosphatases.

Page 29: In the legend to Fig. 1 the term “agarose labelled methylated or un-methylated C69” is used. Do the authors mean “agarose-coupled”?

This mistake is now corrected.

What is the control in Fig. 3E?

We apologize for the over-sight. The control is *Gapdh* and this information is now included in the Methods section.

In Fig. 3C, why was svRNA1 expression not tested?

Our focus was on svRNA4 because it was the only svRNA whose length coincided with m⁵C69. SvRNA1 and 4 should be mutually exclusive, yet the abundance differences of svRNA1 were often not statistically different (e.g. in Fig. 3G). SvRNA1 may be more stable than svRNA4 and therefore more difficult to detect.

In Fig. 5A, B, C: cytoplasm is misspelled.

This mistake is now corrected.

Page 6, line 139, page 11, lines 257,258: change SFSR2 to SRSF2.

This mistake is now corrected.

References:

- Blanco, S., Bandiera, R., Popis, M., Hussain, S., Lombard, P., Aleksic, J., Sajini, A., Tanna, H., Cortes-Garrido, R., Gkatza, N., et al. (2016). Stem cell function and stress response are controlled by protein synthesis. *Nature* 534, 335-340.
- Blanco, S., Dietmann, S., Flores, J.V., Hussain, S., Kutter, C., Humphreys, P., Lukk, M., Lombard, P., Treps, L., Popis, M., et al. (2014). Aberrant methylation of tRNAs links cellular stress to neuro-developmental disorders. *The EMBO journal* 33, 2020-2039.
- Frye, M., Dragoni, I., Chin, S.F., Spiteri, I., Kurowski, A., Provenzano, E., Green, A., Ellis, I.O., Grimmer, D., Teschendorff, A., et al. (2009). Genomic gain of 5p15 leads to over-expression of Misu (NSUN2) in breast cancer. *Cancer Lett.*
- Hussain, S., Sajini, A.A., Blanco, S., Dietmann, S., Lombard, P., Sugimoto, Y., Paramor, M., Gleeson, J.G., Odom, D.T., Ule, J., et al. (2013). NSun2-mediated cytosine-5 methylation of vault noncoding RNA determines its processing into regulatory small RNAs. *Cell reports* 4, 255-261.

Martinez, F.J., Lee, J.H., Lee, J.E., Blanco, S., Nickerson, E., Gabriel, S., Frye, M., Al-Gazali, L., and Gleeson, J.G. (2012). Whole exome sequencing identifies a splicing mutation in NSUN2 as a cause of a Dubowitz-like syndrome. *J Med Genet* 49, 380-385.

Watt, F.M. (1989). Terminal differentiation of epidermal keratinocytes. *Curr Opin Cell Biol* 1, 1107-1115.

Reviewers' comments:

Reviewer #1 (Remarks to the Author):

All of this reviewer's concerns have been addressed.

This referee recommends publication after some important minor issues have been addressed that pertain to the descriptions in the material and methods sections.

PAGE 19:

>> the plasmid from Addgene is called "pLKO.1 puro" not "PLK-puro".

PAGE 22:

>> "adipic acid dihydrazide-agarose" instead of "adipic acid dehydrazide-agarose"

PAGE 22:

>> instead of "5 ul RNaseOUT", please, correctly state the units of RNAsOUT per ml extract rather than just a volume of solution.

PAGE 23:

>> Please, specify how much RNase inhibitor was added to the RNA Immunoprecipitation reactions. This important because any synthetic RNA will be digested by endogenous RNases in the extract during the incubation.

PAGE 25:

>> Please, specify how much RNase was added to the RNA cleavage assay reactions.

>> Please, specify the end identities of the synthetic vtRNAs? Were they protected by 5' or/and 3' groups, which would block exonucleases?

PAGE 26:

>> "All samples were gated using forward versus side scatter." is mentioned twice in the same paragraph.

PAGE 26:

>> "To complete the mapping, tRNA gene predictions were obtained from GtRNadb (<http://lowelab.ucsc.edu/GtRNadb>)." This was copied from the methods section in PMID: 25063673. Since this manuscript does not analyse tRNA methylation this should be clarified.

PAGE 27:

>> "...the heatmaps were reported relative to the annotated transcriptional start sites of the documented beginning of each tRNA." This was copied from the methods section in PMID: 25063673. Since the analysis does not address tRNAs but vtRNAs this should be clarified.

PAGE 27:

>> Interestingly, the sequencing data (GSE122600 and 125046) are not available as specified, and should be made available, if not the referees at time of peer-review, then at least at time of acceptance of the manuscript.

PAGE 27:

>> "Codes for the small RNA sequencing and RNA bisulfite sequencing analyses will be made available on GitHub." sounds great but in light of the missing primary data at GEO, this reviewer urges the authors to keep their word and publish the analysis pipeline.

Reviewer #2 (Remarks to the Author):

In the revised version of their manuscript Sajini et al. addressed and clarified some of the issues raised by the reviewers and they included some new data intended to strengthen the conclusions of the manuscript. However, there are some instances where the reviewer's comments were not addressed adequately and where modifications to the text did not improve clarity. This needs to be fixed before acceptance of the manuscript.

1. Fig. 1: The expression data shown now in Fig. 1H indicate strong overexpression of the transfected constructs. Nevertheless, svRNA4 production in -/- (wt) cells is only slightly increased over -/- cells and remains strongly below that in +/- cells. The authors should give an explanation in the manuscript text as to why overexpression of Nsun2 fails to rescue svRNA4 processing.

2. Fig. 2C: While the authors explained in the rebuttal letter how the quantification of the western blot signals was performed, they do not address the problem of the obviously saturated signal of hnRNPA1, which renders the quantification of the pull-down experiment questionable and does not justify the statement on page 7, line 153: "...and found that SRSF2 bound non-methylated (C69) with higher affinity than methylated (mC69) VTRNA1.1 (Fig. 2C)."

3. Fig. 3AB and page 8, line 180: " First, we confirmed comparable protein expression levels of SRSF2 in NSUN2-expressing (+/+; +/-) or -lacking (-/-) cells by immunoprecipitation and Western blotting (Fig. 3A, B; Fig. S3A). " The text should be changed toimmunoprecipitation OR Western blotting. Otherwise, it suggests that the Western blots shown in Fig. 3A and B are of immunoprecipitated samples.

4. Fig. 3D: The authors fixed the problem of comparing svRNA1 and 4 abundance between nuclear and cytoplasmic extracts. However, the figure is still problematic because qPCR results obtained with different primer pairs are compared. For an accurate quantification of svRNA1 and 4 levels, either Northern blotting or an absolute qPCR quantification using suitable DNA standards are needed.

5. Page 9, line 197: "...The svRNA4 production was only slightly enhanced in presence of nuclear extract when methylated VTRNA1.1 was used (Fig. S3C)". The text should clarify what the comparison is, i.e. enhanced compared to what? Assuming that the comparison between svRNA1 and 4 can be done that way (see point 4), the methylated RNA seems to be a worse substrate for processing than the unmethylated RNA. Does this mean that only unmethylated vtRNA1.1 is cleaved when it is not bound by SRSF2? Does methylation also inhibit cleavage?

6. Fig. 3E: It is not clear, how the numbers were calculated. Moreover, since they seem to be based on the real time PCR results from Fig. 3D the concerns raised in point 4 pertain also to this figure. I suggest to delete this panel.

7. Page 9, line 206: The reference should be to Fig. 1G instead of 1D.

8. Page 11, lines 253-259. It is interesting that methylation and transcript levels of vtRNA1.1 increase during differentiation while NSUN2 and SRSF2 levels decrease. Might this support a possible cleavage-inhibitory role of methylation that is independent of SRSF2? (see point 5).

9. Page 12, line 269: "To provide direct evidence that both the presence of NSUN2 and the RNA processing machinery were required..." Since the experiments examine only the role of the processing product svRNA4 but not that of NSUN2 or the processing machinery, this sentence should be changed.

10. page 12, line 284: Why should the transfection of an svRNA4 mimic repress vtRNA1.1 as stated in the text?

11. Page 14, line 321: Reference to Fig. 5I should be corrected to Fig. 5H.

12. Page 14, line 325: This statement is not entirely correct, because apparently SRSF2 is only required for VTRNA1.1 processing when NSUN2 is perturbed. When only SRSF2 is removed, the authors claim that svRNA4 production is fine. However, as noted in the first review of this paper, this piece of data is not convincing, since one of the 4 data points in the Srsf2 column in Fig. 5H looks very much like an outlier. This needs to be resolved.

13. Page 15, line 368: "...by sequestering un-methylated VTRNA1.1 in the nucleus and thereby protecting it from methylation by NSUN2 (Fig. 6A)." Why should sequestering VTRNA1.1 to the nucleus protect it from methylation by NSUN2, when NSUN2 is mostly present in the nucleus?

NCOMMS-18-02777A

In response to the referees' comments, we have made the following changes to the manuscript:

Reviewer #1:

All of this reviewer's concerns have been addressed.

This referee recommends publication after some important minor issues have been addressed that pertain to the descriptions in the material and methods sections.

PAGE 19:

>> *the plasmid from Addgene is called "pLKO.1 puro" not "PLK-puro".*

This mistake is now corrected

PAGE 22:

>> *"adipic acid dihydrazide-agarose" instead of "adipic acid dehydrazide-agarose"*

This mistake is now corrected

PAGE 22:

>> *instead of "5 ul RNaseOUT", please, correctly state the units of RNAsOUT per ml extract rather than just a volume of solution.*

As the referee requested, we now state that we used 200 units / ml in the Methods section.

PAGE 23:

>> *Please, specify how much RNase inhibitor was added to the RNA Immunoprecipitation reactions. This important because any synthetic RNA will be digested by endogenous RNases in the extract during the incubation.*

The information that 200 units per ml of RNase inhibitor were added to the RNA immunoprecipitation reactions is now included in the Methods section.

PAGE 25:

>> *Please, specify how much RNase was added to the RNA cleavage assay reactions.*
>> *Please, specify the end identities of the synthetic vtRNAs? Were they protected by 5' or/and 3' groups, which would block exonucleases?*

As requested by referee 2, the RNA cleavage assay data have been removed from the manuscript.

PAGE 26:

>> *“All samples were gated using forward versus side scatter.” is mentioned twice in the same paragraph.*

This mistake is now corrected

PAGE 26:

>> *“To complete the mapping, tRNA gene predictions were obtained from GtRNAdb (<http://lowelab.ucsc.edu/GtRNAdb>).” This was copied from the methods section in PMID: 25063673. Since this manuscript does not analyse tRNA methylation this should be clarified.*

We apologize for the oversight and have re-written the Methods section.

PAGE 27:

>> *“...the heatmaps were reported relative to the annotated transcriptional start sites of the documented beginning of each tRNA.” This was copied from the methods section in PMID: 25063673. Since the analysis does not address tRNAs but vtRNAs this should be clarified.*

Please note that we also use tRNA methylation examples in Figure 4F and S4B. To avoid confusion, we have re-written the Methods section.

PAGE 27:

>> *Interestingly, the sequencing data (GSE122600 and 125046) are not available as specified, and should be made available, if not the referees at time of peer-review, then at least at time of acceptance of the manuscript.*

We are unsure why the referee was unable to access the data under the provided access code as stated on the title page: Sequencing data are up-loaded onto dbGaP (phs000645.v4.p1) and GEO (GSE122600, GSE125046)

<https://www.ncbi.nlm.nih.gov/geo/query/acc.cgi?acc=GSE122600>

Reviewer access token: ktgpskayldcpjor

<https://www.ncbi.nlm.nih.gov/geo/query/acc.cgi?acc=GSE125046>

sharscawrhqxhr

We double checked the links and they work. The data will be released upon acceptance of our manuscript.

PAGE 27:

>> *“Codes for the small RNA sequencing and RNA bisulfite sequencing analyses will be made available on GitHub.” sounds great but in light of the missing primary data at GEO, this reviewer urges the authors to keep their word and publish the analysis pipeline.*

Reviewer #2:

In the revised version of their manuscript Sajini et al. addressed and clarified some of the issues raised by the reviewers and they included some new data intended to strengthen the conclusions of the manuscript. However, there are some instances where the reviewer's comments were not addressed adequately and where modifications to the text did not improve clarity. This needs to be fixed before acceptance of the manuscript.

1. *Fig. 1: The expression data shown now in Fig. 1H indicate strong overexpression of the transfected constructs. Nevertheless, svRNA4 production in -/- (wt) cells is only slightly increased over -/- cells and remains strongly below that in +/- cells. The authors should give an explanation in the manuscript text as to why overexpression of Nsun2 fails to rescue svRNA4 processing.*

We disagree with the statement that over-expression of NSUN2 fails to rescue svRNA4-production. Re-expression of the wild-type NSUN2 protein in *NSUN2*^{-/-} cells significantly increases svRNA4-production when compared to the same cells over-expressing to enzymatic dead versions of the protein.

Thus, our experiment not only shows that NSUN2 re-expression rescues formation of svRNA4, it also shows that this was directly dependent on its methylation activity. We believe that it is rather remarkable that re-establishing a single m⁵C site in VTRNA1.1 is sufficient to enhance svRNA4-production.

Regarding the production of svRNA4 in *NSUN2*^{+/-} cells, we would argue that *NSUN2*^{+/-} cells produce exceptional high levels of svRNA4. This cell line was derived from the mother of the two patients who did not present with any disease symptoms (Martinez et al., 2012). Thus, *NSUN2*^{+/-} is an independent cell line that may very well have compensated for the loss of one allele of NSUN2, for instance through the adaptation of the RNA processing machineries.

2. *Fig. 2C: While the authors explained in the rebuttal letter how the quantification of the western blot signals was performed, they do not address the problem of the obviously saturated signal of hnRNPA1, which renders the quantification of the pull-down experiment questionable and does not justify the statement on page 7, line 153: "...and found that SRSF2 bound non-methylated (C69) with higher affinity than methylated (mC69) VTRNA1.1 (Fig. 2C)."*

We are unsure why the referee believes that the signal of the loading control hnRNPA1 was saturated when the very faint unspecific band above hnRNPA1 also demonstrates even loading of the samples. However, to address the referee's concern, we now also included the Coomassie stain of this specific pull-down experiment (see Figure 2c). The Coomassie stain shows very few abundant RNA-binding proteins enriched in the m⁵C69 VTRNA1.1 pull-down when compared to non-methylated C69, thus confirming the mass spectrometry results.

In addition, in the manuscript we provide three independent methods to prove that SRSF2 binds more efficiently to unmethylated VTRNA1.1 (quantitative SILAC mass spectrometry, RNA pull-down and EMSA).

3. Fig. 3AB and page 8, line 180: "First, we confirmed comparable protein expression levels of SRSF2 in NSUN2-expressing (+/+; +/-) or -lacking (-/-) cells by immunoprecipitation and Western blotting (Fig. 3A, B; Fig. S3A). " The text should be changed toimmunoprecipitation OR Western blotting. Otherwise, it suggests that the Western blots shown in Fig. 3A and B are of immunoprecipitated samples.

We agree with the referee and have changed the text as requested.

4. Fig. 3D: The authors fixed the problem of comparing svRNA1 and 4 abundance between nuclear and cytoplasmic extracts. However, the figure is still problematic because qPCR results obtained with different primer pairs are compared. For an accurate quantification of svRNA1 and 4 levels, either Northern blotting or an absolute qPCR quantification using suitable DNA standards are needed.

We apologize that our revised manuscript still lacked clarity on the cleavage assay. The point of this experiment was to test whether the RNA processing machinery involved in generating svRNA4 was located to the nucleus or the cytoplasm. Since the cellular location of VTRNA1.1 processing is not crucial for the main message of our manuscript, we have taken it experiment out of our study (as suggested by the referee under point 6).

5. Page 9, line 197: "...The svRNA4 production was only slightly enhanced in presence of nuclear extract when methylated VTRNA1.1 was used (Fig. S3C)". The text should clarify what the comparison is, i.e. enhanced compared to what? Assuming that the comparison between svRNA1 and 4 can be done that way (see point 4), the methylated RNA seems to be a worse substrate for processing than the unmethylated RNA. Does this mean that only unmethylated vtRNA1.1 is cleaved when it is not bound by SRSF2? Does methylation also inhibit cleavage?

See our response to point 4. We omitted the experiment from our study as it was not essential for the main message of our manuscript.

6. Fig. 3E: It is not clear, how the numbers were calculated. Moreover, since they seem to be based on the real time PCR results from Fig. 3D the concerns raised in point 4 pertain also to this figure. I suggest to delete this panel.

We agree with the referee and have omitted this experiment from our study.

6. Page 9, line 206: The reference should be to Fig. 1G instead of 1D.

This mistake has now been corrected.

8. Page 11, lines 253-259. *It is interesting that methylation and transcript levels of vtRNA1.1 increase during differentiation while NSUN2 and SRSF2 levels decrease. Might this support a possible cleavage-inhibitory role of methylation that is independent of SRSF2? (see point 5).*

This is an excellent suggestion. The data indicate that m⁵C69 might have a cleavage-inhibitory role in differentiating cells leading to the stabilization of the non-coding RNA. Alternatively, the RNA processing machinery could be absent in differentiated cells. We are considering these possibilities for our further studies on this issue.

9. Page 12, line 269: *“To provide direct evidence that both the presence of NSUN2 and the RNA processing machinery were required....” Since the experiments examine only the role of the processing product svRNA4 but not that of NSUN2 or the processing machinery, this sentence should be changed.*

We agree with the referee and have changed the text to *“To provide direct evidence that both the presence of NSUN2 and VTRNA1.1 processing was required ...”*.

10. page 12, line 284: *Why should the transfection of an svRNA4 mimic repress vtRNA1.1 as stated in the text?*

The sequence of svRNA4 is also an antisense of the full-length VT-RNA1.1. A similar inhibitory effect has been described for svRNAs derived from VTRNA2.1 (Minones-Moyano et al., 2013).

11. Page 14, line 321: *Reference to Fig. 5I should be corrected to Fig. 5H.*

This mistake has now been corrected.

12. Page 14, line 325: *This statement is not entirely correct, because apparently SRSF2 is only required for VTRNA1.1 processing when NSUN2 is perturbed. When only SRSF2 is removed, the authors claim that svRNA4 production is fine. However, as noted in the first review of this paper, this piece of data is not convincing, since one of the 4 data points in the Srsf2 column in Fig. 5H looks very much like an outlier. This needs to be resolved.*

We agree with the referee and have omitted the *“only”* from this sentence.

The point of Figure 5H was to test whether the reduction of svRNA4 in the absence of NSUN2 can be rescued by simultaneously knocking-down of *Srsf2*, and this was indeed the case.

Please note that repression of *Srsf2* in primary human keratinocytes alone will be sufficient to indirectly repress NSUN2 expression because it induces differentiation (Fig. 4a-b; Fig. 5d). Therefore, the referee is correct that the one data point may be an outlier. However, it could also reflect the biological variability of the experiment, and since it was not statistically significant, we cannot take any further conclusions from the experiment.

13. Page 15, line 368: "...by sequestering un-methylated VTRNA1.1 in the nucleus and thereby protecting it from methylation by NSUN2 (Fig. 6A)." Why should sequestering VTRNA1.1 to the nucleus protect it from methylation by NSUN2, when NSUN2 is mostly present in the nucleus?

We agree with the referee and have changed this sentence to "*SRSF2 contributes to VTRNA1.1 processing into svRNA4 by binding to un-methylated VTRNA1.1 and thereby protecting it from methylation by NSUN2 (Fig. 6a).*"

References:

Martinez, F.J., Lee, J.H., Lee, J.E., Blanco, S., Nickerson, E., Gabriel, S., Frye, M., Al-Gazali, L., and Gleeson, J.G. (2012). Whole exome sequencing identifies a splicing mutation in NSUN2 as a cause of a Dubowitz-like syndrome. *J Med Genet* 49, 380-385.

Minones-Moyano, E., Friedlander, M.R., Pallares, J., Kagerbauer, B., Porta, S., Escaramis, G., Ferrer, I., Estivill, X., and Marti, E. (2013). Upregulation of a small vault RNA (svtRNA2-1a) is an early event in Parkinson disease and induces neuronal dysfunction. *RNA Biol* 10, 1093-1106.

Responses to the referees

25th March 2019

NCOMMS-18-02777B

There were no more concerns to address.